# Bridging Draft Policy Misalignment: Group Tree Optimization for Speculative Decoding

**Shijing Hu**[1]    **Jingyang Li**[2]    **Zhihui Lu**[1]*    **Pan Zhou**[3]

[1]Fudan University    [2]National University of Singapore    [3] Singapore Management University

sjhu24@m.fudan.edu.cn    li_jingyang@u.nus.edu    lzh@fudan.edu.cn
panzhou@smu.edu.sg

## Abstract

Speculative decoding accelerates large language model (LLM) inference by letting a lightweight draft model propose multiple tokens that the target model verifies in parallel. Yet existing training objectives optimize only a single greedy draft path, while decoding follows a *tree* policy that re-ranks and verifies multiple branches. This *draft policy misalignment* limits achievable speedups. We introduce **Group Tree Optimization** (GTO), which aligns training with the decoding-time tree policy through two components: (i) *Draft Tree Reward*, a sampling-free objective equal to the expected acceptance length of the draft tree under the target model, directly measuring decoding performance; (ii) *Group-based Draft Policy Training*, a stable optimization scheme that contrasts trees from the current and a frozen reference draft model, forming debiased group-standardized advantages and applying a PPO-style surrogate along the longest accepted sequence for robust updates. We further prove that increasing our Draft Tree Reward provably improves acceptance length and speedup. Across dialogue (MT-Bench), code (HumanEval), and math (GSM8K), and multiple LLMs (e.g., LLaMA-3.1-8B, LLaMA-3.3-70B, Vicuna-1.3-13B, DeepSeek-R1-Distill-LLaMA-8B, Qwen3-8B), GTO increases acceptance length by $7.4\%$ and yields an additional $7.7\%$ speedup over prior state-of-the-art EAGLE-3. By *bridging draft policy misalignment*, GTO offers a practical, general solution for efficient LLM inference. Code and draft models are available at https://github.com/hsj576/GTO.

## 1 Introduction

Large language models (LLMs) like GPTs (Achiam et al., 2023) and LLaMAs (Touvron et al., 2023a;b; Dubey et al., 2024) have achieved remarkable success in dialogue (Zheng et al., 2023), coding (Chen et al., 2021), and reasoning (Cobbe et al., 2021). Yet their standard autoregressive decoding remains inefficient: each token requires a full forward pass, making inference both compute-intensive and latency-bound. Speculative decoding (Leviathan et al., 2023; Chen et al., 2023a) mitigates this by introducing a lightweight draft model to propose multiple tokens, which the target LLM verifies in parallel. This enables multi-token generation per target step, substantially reducing inference time.

Recent work has improved speculative decoding by refining draft model training. For instance, HASS (Zhang et al., 2024) enforces feature consistency to reduce hidden-state mismatches, GRIF-FIN (Hu et al., 2025) resolves token-level misalignments, and EAGLE-3 (Li et al., 2025) incorporates training-time rollouts to better mimic decoding. However, they face a fundamental limitation yet: **draft policy misalignment between training and decoding**. That is, the training objective of draft model does not align with how draft sequences are actually generated and used during decoding, ultimately weakening the effectiveness of training for improving decoding performance.

Specifically, during training, given a context, the draft model is optimized to maximize the likelihood of generating the same token as the target model (Li et al., 2024a;b; 2025; Zhang et al., 2024). It treats drafting as a *single-path sequence prediction problem*, and its corresponding optimal *training-time draft policy* is a greedy drafting: select the highest-probability token at each draft step to form a single draft sequence (e.g., the leftmost draft path in Fig. 1 (a)). However, the practice decoding differs from greedy drafting, and indeed adopts *tree drafting* (Li et al., 2024b): as shown in Fig. 1 (a), it uses draft model to expand a draft tree containing multiple draft sequences, then re-ranks

---
*Corresponding author.

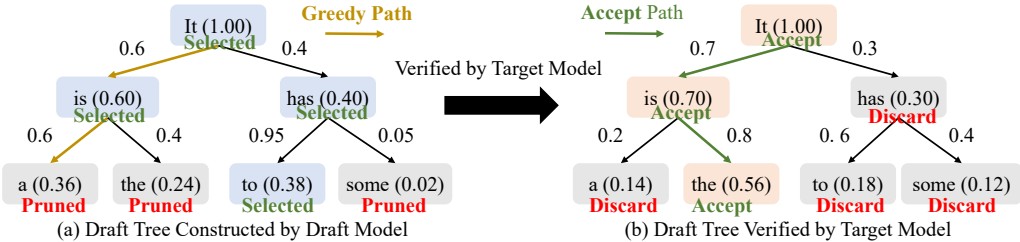

Figure 1: Draft policy misalignment between training and decoding. **(a)** The tree is built by draft model at decoding: number on edge is the token probability predicted by draft model, e.g., "is" (0.6), and number in parentheses is current path confidence, e.g., "It is" ($0.6=1.0 \times 0.6$). Training enforces a training-time greedy draft policy, following the locally best child and yielding the path "It → is → a" (confidence 0.36). At decoding, top-4 re-ranking compares sibling paths, where "It → has → to" (0.38) outperforms the greedy branch which is thus pruned (red). Training signal concentrated on a single greedy path is wasted when sibling branches win. **(b)** Target model verifies the tree with its own probabilities. It compares the confidence of each sequence, and accepts the sequence "It → is → the". Even when the greedy branch survives, target model may accept a different sibling.

sequences using prediction confidences, and finally selects top-$g$ tokens which are then verified by the target LLM. This decoding-time policy is fundamentally different: unlike the training-time policy focusing on a single greedy draft path (the most left one in Fig. 1 (a)), it leverages multiple high-quality branches (the whole tree in Fig. 1 (a)) to maximize the expected acceptance length.

This draft policy misalignment leads to two characteristic failure modes: (1) greedy path pruning; and (2) verification mismatch. For (1), due to re-ranking and top-$g$ selection, the optimal training-time greedy path may be pruned at decoding if sibling branches achieve higher overall confidence. For example, in Fig. 1(a), the greedy sequence "It is a" (confidence 0.36) is discarded in favor of the sibling "It has to" (confidence 0.38). Regarding (2), even when the greedy path survives pruning, target model may accept a different branch, e.g., accepting "It is the" rather than the greedy "It is a" in Fig. 1(b). In both cases, training effort spent on the greedy path yields little decoding benefit. These failures also reveal a structural bottleneck: training encourages convergence to a policy that is effective and optimal only under single-path greedy drafting, but suboptimal for the tree-based strategy used in practice, causing training-decoding misalignment and limiting decoding efficiency. Bridging this gap is therefore crucial for realizing the full potential of speculative decoding in LLMs.

We empirically validate this misalignment using the EAGLE-3 draft model on LLaMA-3.1-8B (Dubey et al., 2024). As shown in Fig. 2(a), 19–34% of greedy paths are pruned during draft tree construction, and the finally accepted path matches the greedy one only 36–49% of the time. Even when accepted, the greedy path averages $3-4$ tokens, shorter than the $5-6$ tokens of the full tree (Fig. 2(b)). This confirms that greedy training overlooks globally optimal sequences, highlighting the severity of draft policy misalignment and its direct impact on speculative decoding efficiency.

**Contributions:** To address the draft policy misalignment, we propose Group Tree Optimization (GTO), a novel training algorithm for speculative decoding that explicitly optimizes the tree-based draft policy rather than a single greedy path. By aligning training with the actual decoding procedure, GTO ensures that draft models learn policies that directly improve decoding-time efficiency.

First, we introduce a draft-tree reward that directly aligns training with the decoding-time policy. Unlike prior methods that optimize token-level accuracy (Li et al., 2025; Hu et al., 2025; Zhang et al., 2024), GTO adopts the same rollout strategy used during decoding: the draft model generates a tree of candidate sequences, which is then verified by the target LLM. We define the reward as the *expected acceptance length* of the tree, a direct measure of decoding efficiency. This shifts the objective from "predicting the next token correctly" to "producing draft trees that survive verification and extend accepted prefixes as far as possible," aligning the training goal with real decoding.

Second, we develop a stable and effective draft policy training algorithm to maximize this draft-tree reward and thus boost decoding efficiency. Training is challenging because rewards are sparse, position-dependent, and high variance. GTO addresses this with a group-based approach tailored to deterministic draft-tree rollouts. We sample small groups of trees under both the current draft model and a frozen reference, and use their contrasts to construct debiased tree-level rewards that cancel

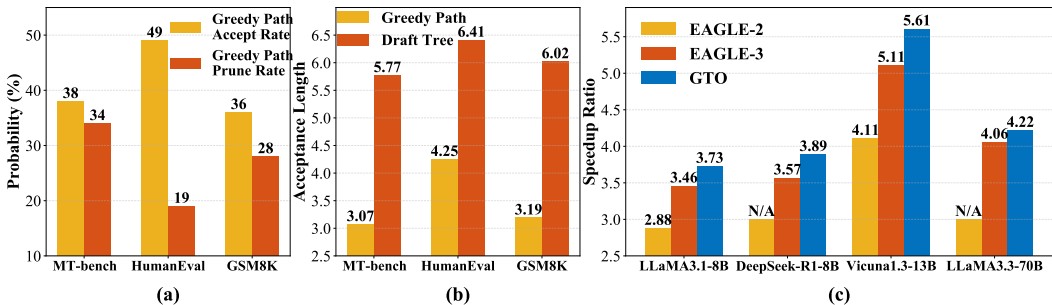

Figure 2: Experimental Results of Draft Policy Misalignment between Training and Decoding. (a) Fraction of training-time greedy paths that are *pruned* during draft tree construction (orange bars) and fraction where the *accepted path coincides* the greedy path (yellow bars). (b) Accepted greedy paths are also *shorter*: their average acceptance length is $3-4$ tokens, compared to $5-6$ for the entire draft tree. (c) Speedup Ratio Comparison of GTO and EAGLE-3.

position-specific difficulty. Within each group, standardized advantages normalize rewards across contexts, reducing variance and improving credit assignment by highlighting which branches truly drive longer accepted prefixes. Finally, we optimize a PPO-style clipped objective, defined over the likelihood ratio along the longest accepted sequence, ensuring robust and efficient training.

Finally, we validate GTO across dialogue (MT-Bench (Zheng et al., 2023)), code (HumanEval (Chen et al., 2021)), and reasoning (GSM8K (Cobbe et al., 2021)) benchmarks on LLaMA-3.1-8B, LLaMA-3.3-70B, DeepSeek-R1-Distill-LLaMA-8B, and Vicuna-13B. GTO consistently improves acceptance length by $7.4\%$ over EAGLE-3, translating into an additional $7.7\%$ speedup (Fig. 2 (c)).

## 2 RELATED WORK

Speculative decoding accelerates LLM inference by splitting each step into a lightweight *draft* and a *verification* stage (Sun et al., 2024; Miao et al., 2024; Chen et al., 2023b; Kim et al., 2024; Liu et al., 2023). Existing methods vary in how drafts are produced and verified: prompt- and retrieval-based approaches (PLD, Lookahead, CLLMs) improve draft quality but degrade with scarce context (Saxena, 2023; Fu et al., 2024; Kou et al., 2024); tree-based verification (Sequoia, SpecExec) boosts acceptance but often increases compute (Chen et al., 2024; Svirschevski et al., 2024); REST and Ouroboros reuse outputs or databases but depend on resource quality (He et al., 2023; Zhao et al., 2024); hybrid designs (Chimera, Glide) partially integrate the target model at extra cost (Zeng et al., 2024; Du et al., 2024). Efficiency-oriented drafters span Medusa, Hydra, and RNN/Transformer-based models such as EAGLE-3, with methods like HASS and GRIFFIN addressing feature- and token-level mismatches (Cai et al., 2024; Ankner et al., 2024; Cheng et al., 2024; Li et al., 2024a;b; Zhang et al., 2024; Hu et al., 2025; Li et al., 2025). Despite these advances, a key limitation remains: *draft policy misalignment*, where training optimizes a single greedy path but decoding verifies a *tree* of candidates. We propose GTO to align the training objective with the decoding-time tree policy, improving acceptance length and speedup. GTO complements existing methods and provides a general solution to policy mismatch in speculative decoding.

## 3 GTO: GROUP TREE OPTIMIZATION

To address the *draft policy misalignment* highlighted in Section 1, we introduce *Group Tree Optimization* (GTO), a training framework that explicitly aligns the draft policy with decoding. The central idea is to evaluate and optimize draft policies not on a single greedy path, but on entire draft *trees*, using the same drafting procedure deployed at decoding. To this end, GTO consists of two key components: (i) a *draft-tree reward* that faithfully measures expected decoding performance in terms of accepted draft sequence length (Section 3.1), and (ii) a stable group-based optimization algorithm for training with this reward (Section 3.2). Below we introduce them in turn.

### 3.1 DRAFT TREE REWARD

The effectiveness of speculative decoding is governed by the length of accepted draft sequence: the longer the draft sequence accepted by the target model, the fewer verification steps are needed, and thus the greater the decoding efficiency. With the same draft model, a higher expected acceptance

length directly translates to higher speedup. This makes *expected acceptance length* the most faithful measure of practical decoding performance when using the same draft model.

To capture this, GTO eliminates the traditional mismatch between training and decoding: instead of optimizing token-level proxies along a greedy path, we construct draft *trees* during training using the same decoding-time expansion and pruning policy (e.g., EAGLE-2–style multi-branch expansion, reranking and selection). The draft model is then optimized with respect to a *tree-level reward* that directly reflects its expected decoding-time utility.

Formally, given a training prefix (a.k.a., context) $\mathbf{x}_{1:t}$, we follow EAGLE-2, and construct a depth-$d$ draft tree $\mathbf{T}_t$ with the draft model $\mathcal{M}$:

$$\mathbf{T}_t = \mathcal{G}(\mathcal{M}, \mathbf{x}_{1:t}), \tag{1}$$

where $\mathcal{G}$ denotes the decoding policy. The policy $\mathcal{G}$ grows the tree in two stages.

*(i) Layer-wise expansion.* At depth $\ell \in \{1, \ldots, d\}$, consider all frontier expansions (token edges) from the current layer. For each candidate expansion we compute a global acceptance score. We then select the top-$k$ token expansions across the entire layer according to the global acceptance score and expand draft tree only on these children. This global competition allows promising siblings to outcompete locally greedy choices and prevents early commitment to a single path.

*(ii) Global pruning and re-ranking.* After reaching the maximum depth, we collect all leaves and re-rank them by the global acceptance score. We retain the top-$g$ leaves and prune the rest.

The tree consists of $N$ candidate sequences $\mathbf{T}_t = \{\mathbf{S}_{t,1}, \ldots, \mathbf{S}_{t,N}\}$, each of length $l_i \leq d$ may be different due to selection (pruning):

$$\mathbf{S}_{t,i} = \{\bar{\mathbf{x}}_{t+1,i}, \ldots, \bar{\mathbf{x}}_{t+l_i,i}\}, \tag{2}$$

where $\{\bar{\mathbf{x}}_{t+1,i}, \ldots, \bar{\mathbf{x}}_{t+l_i,i}\}$ denotes the draft sequence $\mathbf{S}_{t,i}$. Then, for each sequence, we define its *expected acceptance length* under the target model $\mathcal{T}$:

$$\mathbf{L}_{t,i} = \sum_{j=1}^{l_i} \mathcal{P}(\bar{\mathbf{x}}_{t+j,i} \,|\, \mathbf{x}_{1:t}, \bar{\mathbf{x}}_{t+1:t+j-1,i}), \tag{3}$$

with

$$\mathcal{P}(\bar{\mathbf{x}}_{t+j,i} \,|\, \mathbf{x}_{1:t}, \bar{\mathbf{x}}_{t+1:t+j-1,i}) = \prod_{k=1}^{j} \mathcal{T}(\bar{\mathbf{x}}_{t+k,i} \,|\, \mathbf{x}_{1:t}, \bar{\mathbf{x}}_{t+1:t+k-1,i}). \tag{4}$$

Here, $\mathbf{L}_{t,i}$ is the expectation of how many tokens of $\mathbf{S}_{t,i}$ will be accepted by target model $\mathcal{T}$. This definition is sampling-free, while remaining directly tied to decoding performance.

Accordingly, we can average the expected acceptance length of all sequences in the tree to measure the overall decoding performance of the tree. However, since decoding utility depends on which sequences (branches) survive pruning, we aggregate the sequence-level expectations with a smooth max (log-sum-exp), balancing differentiability with a focus on the strongest sequences:

$$\mathbf{r}_t = \mathcal{R}(\mathbf{T}_t; \eta) = \frac{1}{\eta} \log\left(\sum_{i=1}^{N} \exp(\eta \, \mathbf{L}_{t,i})\right), \tag{5}$$

where the temperature $\eta > 0$ interpolates between the maximum ($\eta \to \infty$) and the average ($\eta \to 0$) branch acceptance length. We set $\eta = 1$ in experiments, which yields a stable and informative reward and works very well in our all experiments. Ablation results in Table 3 show this strategy is better than average all expected length or use the maximum length.

By training the draft model to maximize $\mathcal{R}(\mathbf{T}_t)$, GTO ensures that the draft *policy* and training *objective* are fully aligned with decoding. Unlike prior approaches that rely on token-level log-likelihoods or greedy-path proxies, GTO directly optimizes for the expected acceptance length that governs speculative decoding speedup.

**Theoretical guarantee.** Importantly, improving the Draft Tree Reward provably increases the expected decoding acceptance length, regardless of the target model's sampling temperature:

**Theorem 1** (Maximizing Draft Tree Reward Guarantees Improved Expected Acceptance Length).
*Consider a draft tree $\mathbf{T}_t$ and target model temperature $T \geq 0$. Let $L_T^{\mathrm{dec}}(\mathbf{T}_t)$ denote the expected acceptance length at decoding. Then:*

(a) *For $T > 0$, if the draft tree reward $\mathbf{r}_t$ increases, then $\mathbb{E}[L_T^{\mathrm{dec}}(\mathbf{T}_t)]$ strictly increases.*

(b) *For $T = 0$, if $\mathbf{r}_t$ increases, then $\mathbb{E}[L_0^{\mathrm{dec}}(\mathbf{T}_t)] = \max_i \mathbf{L}_{t,i}$ also increases.*

See its proof in Appendix A. This result establishes *expected acceptance length* as the key link between training and decoding: optimizing the draft-tree reward directly improves speculative decoding efficiency in practice.

## 3.2 TREE REWARD OPTIMIZATION

Directly optimizing the tree-level reward is challenging, particularly early in training when the draft model is weak and draft-token acceptance rates are low. In this regime, the tree reward is small and high-variance, making naive optimization inefficient and unstable. To address this, following LLM's two-phase training (pretraining and fine-tuning), GTO adopts a two-phase group-based approach: an optional warmup to obtain a competent draft model, followed by a structured group-wise optimization that stabilizes and accelerates training. This design improves sample efficiency and can skip the warmup if a strong pretrained draft model is available. For example, in practice, we can directly use the draft model well trained by EAGLE-3, GRIFFIN and HASS as the reference draft model, which plays a role as the Phase I training.

**Phase I: Draft model warmup.** We first train a reference draft model $\mathcal{M}_0$ using standard token-level objectives like the ones in EAGLE-3 and GRIFFIN. This phase provides a baseline model to stabilize subsequent group-based updates and can be skipped when a sufficiently strong draft model exists, e.g., draft model well trained by EAGLE-3 and GRIFFIN.

**Phase II: Group-based optimization of the draft tree reward.** We now optimize the draft tree reward while ensuring stability and robustness. Inspired by group-based reinforcement learning methods (e.g., GRPO (Shao et al., 2024)), we sample groups of related examples and use group-wise advantage estimation to reinforce high-performing samples while suppressing underperforming ones. However, unlike standard RL, for a fixed prefix $\mathbf{x}_{1:t}$ the draft-tree generation $\mathcal{G}(\mathcal{M}, \mathbf{x}_{1:t})$ is effectively deterministic given the policy, limiting the utility of multiple rollouts from the same state. To enable variance reduction and within-context comparisons, we form *groups* from nearby positions in the same sequence and optimize a clipped likelihood-ratio surrogate with group-normalized advantages.

**Grouping.** Let the training sequence be $\mathbf{x}_{1:\mathbf{s}} = (x_1, \ldots, x_{\mathbf{s}})$, where $\mathbf{s}$ is the sequence length. We partition positions into $K$ *non-overlapping* groups of adjacent indices. Each group is defined by a start index $t_k$ and a fixed group size $m$ (with $m \in [4, 8]$ in practice):

$$\mathbf{G}^{(k)} = \{\, t_k, t_k + 1, \ldots, t_k + m - 1 \,\} \subseteq \{1, \ldots, \mathbf{s}\}, \tag{6}$$

subject to

$$1 \leq t_k \leq \mathbf{s} - m + 1, \qquad t_{k+1} \geq t_k + m \quad \text{(non-overlap)}. \tag{7}$$

The number of groups $K$ is determined by the available compute budget and the sequence length (upper bounded by $\lfloor \mathbf{s}/m \rfloor$).

For every position $i \in \mathbf{G}^{(k)}$, we construct a depth-limited draft tree with the current draft model $\mathcal{M}$ using the decoding policy $\mathcal{G}$:

$$\mathbf{T}_i = \mathcal{G}(\mathcal{M}, \mathbf{x}_{1:i}). \tag{8}$$

By construction, indices within a group are adjacent: for any $i, j \in \mathbf{G}^{(k)}$ we have $|i - j| \leq m - 1$. Consequently, the corresponding prefixes $\mathbf{x}_{1:i}$ and $\mathbf{x}_{1:j}$ differ by at most $m - 1$ trailing tokens and share a long common context. Comparing tree-level rewards only *within* a group therefore: (i) matches examples under nearly identical contexts, (ii) reduces variance in reward comparisons caused by position-specific difficulty, and (iii) yields more reliable credit assignment across nearby

prefixes. Intuitively, we aggregate draft trees from adjacent prefixes so that the within-group differences are small, enabling stable and sample-efficient learning signals.

**Reward shaping and standardization.** A key challenge in draft tree reward optimization is that raw tree rewards $\mathcal{R}(\mathbf{T}_i)$ exhibit *systematic difficulty bias*: some prefixes $\mathbf{x}_{1:i}$ are inherently harder to continue than others, leading to lower acceptance rates regardless of draft quality. For instance, prefixes ending with complex mathematical expressions or rare tokens may consistently yield shorter accepted sequences, while simple conversational prefixes may achieve high acceptance even with suboptimal drafts. This bias confounds the learning signal and can cause the model to avoid challenging contexts rather than improving on them.

To remove systematic difficulty bias across prefixes, we construct reference trees $\bar{\mathbf{T}}_i = \mathcal{G}(\mathcal{M}_0, \mathbf{x}_{1:i})$ to debias the tree reward:

$$\mathbf{R}_i = \mathcal{R}(\mathbf{T}_i) - \mathcal{R}(\bar{\mathbf{T}}_i), \tag{9}$$

where $\mathcal{R}$ is the tree-level reward from Section 3.1. Within each group, rewards are standardized to stabilize updates:

$$\mathcal{A}_i = \frac{\mathbf{R}_i - \text{mean}(\{\mathbf{R}_j\}_{j \in \mathbf{G}^{(k)}})}{\text{std}(\{\mathbf{R}_j\}_{j \in \mathbf{G}^{(k)}}) + \delta}, \tag{10}$$

with a small $\delta > 0$ for numerical stability. Our ablation study (Table 5) demonstrates that without debiasing, the model training will becomes unstable due to high variance in gradient magnitudes, leading to worser performance in decoding.

**Clipped likelihood-ratio objective.** Let $\widehat{\mathbf{S}}_i$ be the longest accepted sequence in $\mathbf{T}_i$ under $\mathcal{T}$, with length $l_i$. Define a per-token likelihood ratio (geometric mean) between $\mathcal{M}$ and $\mathcal{M}_0$ on $\widehat{\mathbf{S}}_i$:

$$s_i = \exp\left(\frac{\log \mathcal{M}(\widehat{\mathbf{S}}_i \mid \mathbf{x}_{1:i}) - \log \mathcal{M}_0(\widehat{\mathbf{S}}_i \mid \mathbf{x}_{1:i})}{\max(l_i, 1)}\right). \tag{11}$$

We then optimize a PPO-style clipped surrogate over each group (Schulman et al., 2017):

$$\mathcal{L}_{\text{GTO}} = -\frac{1}{m} \sum_{i \in \mathbf{G}^{(k)}} \min\left(s_i \cdot \mathcal{A}_i, \text{ clip}(s_i, 1 - \epsilon, 1 + \epsilon) \cdot \mathcal{A}_i\right), \tag{12}$$

where $\text{clip}(s, a, b) = \max\{a, \min\{s, b\}\}$ and $\epsilon > 0$ controls update magnitude.

**Overall training objective.** We combine the group-tree objective with a token-level loss $\mathcal{L}_{\text{token}}$ using a scalar weight $\omega$:

$$\mathcal{L} = \mathcal{L}_{\text{token}} + \omega \cdot \mathcal{L}_{\text{GTO}}. \tag{13}$$

$\mathcal{L}_{\text{token}}$ denotes the token-level cross-entropy loss introduced in EAGLE-3 (Li et al., 2025) that matches the draft model $\mathcal{M}$ to the target model $\mathcal{T}$ under the same prefixes.

This two-phase group-based procedure transforms the decoding-faithful draft tree reward into a stable and effective learning signal, enabling the draft model to reliably maximize expected acceptance length and align training with practical decoding performance. Details are summarized in Appendix. B and Algorithm 1.

## 4 EXPERIMENT

**Models & datasets.** We test GTO on a representative set of LLMs, including LLaMA-3.1-Instruct-8B (Touvron et al., 2023b), LLaMA-3.3-Instruct-70B (Touvron et al., 2023b), Vicuna-1.3-13B (Fan et al., 2025), DeepSeek-R1-Distill-LLaMA-8B (Guo et al., 2025) and Qwen3-8B (Yang et al., 2025). All experiments are conducted on a single NVIDIA A100 80GB GPU, except for LLaMA-3.3-70B, which requires two GPUs. We benchmark performance on three widely used evaluation suites: multi-turn conversation (MT-Bench (Zheng et al., 2023)), code generation (HumanEval (Chen et al., 2021)), and mathematical reasoning (GSM8K (Cobbe et al., 2021)).

**Baselines & implementations.** Vanilla autoregressive decoding serves as the baseline (speedup ratio = $1.00\times$). For comparison, we include recent SoTA speculative decoding methods: SPS (with Vicuna-68M as draft) (Leviathan et al., 2023), PLD (Saxena, 2023), Lookahead (Fu et al., 2024), Medusa (Cai et al., 2024), EAGLE (Li et al., 2024a), EAGLE-2 (Li et al., 2024b), HASS (Zhang

Table 1: Comparison of speedup ratio $SR$ and acceptance length $\tau$ on standard LLM benchmarks with temperature $T \in \{0, 1\}$.

| Model | Method | Temperature = 0 | | | | | | | | Temperature = 1 | | | | | | | |
|---|---|---|---|---|---|---|---|---|---|---|---|---|---|---|---|---|---|
| | | MT-bench | | HumanEval | | GSM8K | | Average | | MT-bench | | HumanEval | | GSM8K | | Average | |
| | | $SR\uparrow$ | $\tau\uparrow$ | $SR\uparrow$ | $\tau\uparrow$ | $SR\uparrow$ | $\tau\uparrow$ | $SR\uparrow$ | $\tau\uparrow$ | $SR\uparrow$ | $\tau\uparrow$ | $SR\uparrow$ | $\tau\uparrow$ | $SR\uparrow$ | $\tau\uparrow$ | $SR\uparrow$ | $\tau\uparrow$ |
| LLaMA-3.1 Instruct 8B | PLD | 1.53 | 1.61 | 1.69 | 1.73 | 1.79 | 1.85 | 1.67 | 1.73 | N/A, since the acceptance conditions are relaxed | | | | | | | |
| | Lookahead | 1.61 | 1.67 | 1.72 | 1.78 | 1.84 | 1.93 | 1.72 | 1.79 | | | | | | | | |
| | EAGLE | 1.73 | 2.97 | 2.43 | 3.26 | 2.04 | 3.06 | 2.07 | 3.10 | 1.62 | 2.39 | 1.97 | 3.08 | 1.92 | 2.89 | 1.84 | 2.79 |
| | EAGLE-2 | 2.52 | 4.02 | 3.31 | 4.70 | 2.83 | 4.21 | 2.89 | 4.31 | 2.04 | 3.13 | 2.62 | 4.37 | 2.37 | 3.71 | 2.34 | 3.74 |
| | GRIFFIN | 2.95 | 4.68 | 3.73 | 5.90 | 3.15 | 5.16 | 3.28 | 5.25 | 2.29 | 3.90 | 3.24 | 5.39 | 2.66 | 4.67 | 2.73 | 4.65 |
| | EAGLE-3 | 3.27 | 5.77 | 3.68 | 6.41 | 3.41 | 6.02 | 3.46 | 6.07 | 2.37 | 4.51 | 3.07 | 5.73 | 2.88 | 5.37 | 2.77 | 5.20 |
| | **GTO** | **3.44** | **6.15** | **4.17** | **6.95** | **3.59** | **6.47** | **3.73** | **6.52** | **2.49** | **4.70** | **3.17** | **5.92** | **3.02** | **5.75** | **2.89** | **5.46** |
| Vicuna-1.3 13B | SPS | 1.91 | 2.24 | 2.18 | 2.52 | 1.74 | 2.00 | 1.94 | 2.25 | 1.59 | 1.81 | 1.73 | 1.99 | 1.47 | 1.75 | 1.60 | 1.85 |
| | Medusa | 1.99 | 2.48 | 2.37 | 2.74 | 2.18 | 2.61 | 2.18 | 2.61 | N/A, since the acceptance conditions are relaxed | | | | | | | |
| | Hydra | 2.57 | 3.25 | 3.02 | 3.68 | 2.61 | 3.37 | 2.73 | 3.43 | | | | | | | | |
| | EAGLE | 2.81 | 3.67 | 3.23 | 4.12 | 2.74 | 3.62 | 2.93 | 3.80 | 2.14 | 3.06 | 2.48 | 3.46 | 2.35 | 3.37 | 2.32 | 3.30 |
| | EAGLE2 | 3.79 | 4.78 | 4.71 | 5.37 | 3.83 | 4.72 | 4.11 | 4.96 | 3.47 | 4.33 | 3.84 | 4.87 | 3.15 | 4.36 | 3.49 | 4.52 |
| | EAGLE-3 | 4.84 | 6.59 | 5.61 | 7.33 | 4.87 | 6.48 | 5.11 | 6.80 | 4.03 | 5.64 | 4.61 | 6.36 | 4.16 | 5.79 | 4.27 | 5.93 |
| | **GTO** | **5.23** | **7.01** | **6.06** | **7.95** | **5.55** | **6.92** | **5.61** | **7.29** | **4.10** | **5.71** | **4.77** | **6.52** | **4.90** | **6.05** | **4.59** | **6.09** |
| DeepSeek-R1 Distill-LLaMA 8B | PLD | 1.34 | 1.42 | 1.53 | 1.62 | 1.48 | 1.54 | 1.45 | 1.53 | N/A, since the acceptance conditions are relaxed | | | | | | | |
| | Lookahead | 1.52 | 1.61 | 1.64 | 1.71 | 1.62 | 1.68 | 1.59 | 1.67 | | | | | | | | |
| | GRIFFIN | 2.71 | 4.24 | 3.19 | 5.23 | 3.42 | 5.58 | 3.11 | 5.02 | 2.38 | 3.93 | 2.83 | 4.68 | 3.13 | 5.23 | 2.78 | 4.61 |
| | EAGLE-3 | 3.34 | 5.32 | 3.59 | 5.88 | 3.78 | 6.16 | 3.57 | 5.79 | 2.71 | 4.54 | 3.15 | 5.10 | 3.49 | 5.82 | 3.11 | 5.15 |
| | **GTO** | **3.49** | **5.60** | **3.98** | **6.58** | **4.20** | **6.92** | **3.89** | **6.37** | **2.76** | **4.59** | **3.34** | **5.44** | **3.71** | **6.50** | **3.27** | **5.51** |
| LLaMA-3.3 Instruct 70B | PLD | 1.43 | 1.51 | 1.58 | 1.67 | 1.52 | 1.61 | 1.51 | 1.60 | N/A, since the acceptance conditions are relaxed | | | | | | | |
| | Lookahead | 1.58 | 1.66 | 1.71 | 1.79 | 1.73 | 1.82 | 1.67 | 1.76 | | | | | | | | |
| | EAGLE-3 | 3.78 | 5.40 | 4.41 | 6.26 | 3.99 | 5.90 | 4.06 | 5.85 | 3.68 | 5.18 | 4.05 | 5.85 | 3.88 | 5.65 | 3.87 | 5.56 |
| | **GTO** | **3.97** | **5.56** | **4.68** | **6.51** | **4.11** | **6.25** | **4.22** | **6.14** | **3.90** | **5.34** | **4.21** | **6.20** | **4.07** | **5.82** | **4.06** | **5.78** |
| Qwen-3 8B | EAGLE-3 | 2.37 | 4.53 | 2.56 | 4.88 | 2.59 | 5.02 | 2.50 | 4.81 | 2.24 | 4.23 | 2.32 | 4.77 | 2.51 | 4.84 | 2.36 | 4.61 |
| | **GTO** | **2.49** | **4.84** | **2.78** | **5.37** | **2.88** | **5.51** | **2.72** | **5.24** | **2.29** | **4.37** | **2.49** | **5.03** | **2.63** | **5.17** | **2.47** | **4.86** |

et al., 2024), GRIFFIN (Hu et al., 2025), and EAGLE-3 (Li et al., 2025). Whenever available, we rely on public implementations and strictly reproduce their decoding policies and hyperparameters.

By default, GTO initializes its draft model from the one provided by EAGLE-3. To assess compatibility, we also experiment with draft models trained by other approaches (see Table 2). The initialized draft models are then fine-tuned with GTO on the ShareGPT dataset (Chiang et al., 2023), except for the reasoning model DeepSeek-R1-Distill-LLaMA 8B, which is also fine-tuned on OpenThoughts-114k-math dataset (Guha et al., 2025). See additional training details for GTO in Appendix B, and details for the baselines in Appendix C.

**Metrics.** For fairness and consistency, we follow priors, e.g., HASS, GRIFFIN, and EAGLE-3, and fix the batch size to 1 and evaluate under decoding temperatures $T \in \{0, 1\}$. Same as prior works like EAGLE-3, GTO is lossless and can preserve output quality. Thus, we focus on two efficiency metrics: (i) **Speedup Ratio** ($SR$) — the runtime acceleration relative to vanilla decoding, and (ii) **Acceptance Length** ($\tau$) — the average number of tokens accepted per draft-verification cycle.

## 4.1 MAIN RESULTS

**Comparison with SoTAs.** We report the acceptance lengths ($\tau$) and speedup ratios ($SR$) of GTO and all baselines across three benchmarks in Table 1. One can observe that GTO consistently outperforms all baselines, including SoTA EAGLE-3, across all datasets, models, and temperature settings. On average, each GTO drafting–verification cycle accepts 6–7 tokens, compared to 5–6 tokens for EAGLE-3. As a result, in terms of tangible wall-clock speedups, GTO improves the runner-up EAGLE-3 by 7.7% for temperature zero and 5.6% for temperature one in an average across four evaluation models, while preserving the lossless property of speculative decoding.

Specifically, on the multi-turn conversation benchmark (MT-Bench), GTO achieves steady gains across all models. For example, with LLaMA-3.1 8B at $T=0$, GTO improves the speedup ratio by 5.2% over EAGLE-3, and by 5.1% at $T=1$. Vicuna-1.3 13B shows even larger gains, reaching 8.1% at $T=0$ and 1.7% at $T=1$. For code generation (HumanEval), the improvements are more pronounced. With LLaMA-3.1 8B, GTO yields a 13.3% speedup increase at $T=0$ and 3.3% at $T=1$. DeepSeek-R1 8B follows the same trend, achieving 10.9% and 6.0% improvements at $T=0$

Table 2: Comparison of speedup ratio ($SR$) and acceptance length ($\tau$) when respectively using draft models trained by GRIFFIN and HASS as initialization of GTO.

| Model | Method | Temperature = 0 | | | | | | | | Temperature = 1 | | | | | | | |
|---|---|---|---|---|---|---|---|---|---|---|---|---|---|---|---|---|---|
| | | MT-bench | | HumanEval | | GSM8K | | Average | | MT-bench | | HumanEval | | GSM8K | | Average | |
| | | $SR\uparrow$ | $\tau\uparrow$ | $SR\uparrow$ | $\tau\uparrow$ | $SR\uparrow$ | $\tau\uparrow$ | $SR\uparrow$ | $\tau\uparrow$ | $SR\uparrow$ | $\tau\uparrow$ | $SR\uparrow$ | $\tau\uparrow$ | $SR\uparrow$ | $\tau\uparrow$ | $SR\uparrow$ | $\tau\uparrow$ |
| LLaMA-3 Instruct 8B | GRIFFIN | 3.09 | 4.85 | 3.65 | 5.97 | 3.30 | 5.31 | 3.35 | 5.38 | 2.62 | 4.35 | 3.31 | 5.62 | 3.07 | 5.08 | 3.00 | 5.02 |
| | GTO | **3.28** | **5.17** | **4.03** | **6.44** | **3.53** | **5.73** | **3.61** | **5.78** | **2.74** | **4.54** | **3.47** | **5.95** | **3.23** | **5.41** | **3.15** | **5.30** |
| | HASS | 2.75 | 4.63 | 3.51 | 5.70 | 3.09 | 5.06 | 3.12 | 5.13 | 2.41 | 4.15 | 3.09 | 5.41 | 2.92 | 4.90 | 2.81 | 4.82 |
| | GTO | **2.95** | **4.96** | **3.86** | **6.19** | **3.33** | **5.47** | **3.38** | **5.54** | **2.54** | **4.36** | **3.23** | **5.69** | **3.06** | **5.25** | **2.94** | **5.10** |
| LLaMA-2 Chat 7B | GRIFFIN | 3.12 | 5.11 | 3.61 | 5.93 | 3.10 | 5.27 | 3.28 | 5.44 | 2.81 | 4.81 | 3.33 | 5.63 | 3.06 | 5.26 | 3.07 | 5.23 |
| | GTO | **3.34** | **5.51** | **3.82** | **6.26** | **3.27** | **5.56** | **3.48** | **5.78** | **2.97** | **5.12** | **3.54** | **5.98** | **3.24** | **5.62** | **3.25** | **5.57** |
| | HASS | 2.97 | 4.97 | 3.46 | 5.69 | 3.06 | 5.12 | 3.17 | 5.26 | 2.72 | 4.64 | 3.18 | 5.22 | 2.83 | 5.08 | 2.91 | 4.98 |
| | GTO | **3.13** | **5.15** | **3.64** | **5.95** | **3.15** | **5.31** | **3.31** | **5.47** | **2.84** | **4.82** | **3.41** | **5.69** | **3.09** | **5.34** | **3.11** | **5.28** |

Table 3: Ablation of draft tree reward aggregation on LLaMA-3.1 8B.

| Method | Temperature = 0 | | | | | | | | Temperature = 1 | | | | | | | |
|---|---|---|---|---|---|---|---|---|---|---|---|---|---|---|---|---|
| | MT-bench | | HumanEval | | GSM8K | | Average | | MT-bench | | HumanEval | | GSM8K | | Average | |
| | $SR\uparrow$ | $\tau\uparrow$ | $SR\uparrow$ | $\tau\uparrow$ | $SR\uparrow$ | $\tau\uparrow$ | $SR\uparrow$ | $\tau\uparrow$ | $SR\uparrow$ | $\tau\uparrow$ | $SR\uparrow$ | $\tau\uparrow$ | $SR\uparrow$ | $\tau\uparrow$ | $SR\uparrow$ | $\tau\uparrow$ |
| **GTO (LSE)** | **3.44** | **6.15** | **4.17** | **6.95** | **3.59** | **6.47** | **3.73** | **6.52** | **2.49** | **4.70** | **3.17** | **5.92** | **3.02** | **5.75** | **2.89** | **5.46** |
| Max | 3.38 | 6.05 | 4.06 | 6.80 | 3.52 | 6.36 | 3.65 | 6.40 | 2.46 | 4.65 | 3.12 | 5.84 | 2.97 | 5.66 | 2.85 | 5.38 |
| Sum (Average) | 3.29 | 5.92 | 3.95 | 6.62 | 3.42 | 6.18 | 3.55 | 6.24 | 2.41 | 4.56 | 3.04 | 5.72 | 2.90 | 5.55 | 2.78 | 5.28 |

and $T{=}1$, respectively. These results highlight the effectiveness of GTO's tree-based optimization for structured generation tasks such as coding. On mathematical reasoning (GSM8K), GTO again surpasses EAGLE-3 across all configurations. For instance, with DeepSeek-R1 8B, GTO delivers an 11.1% speedup improvement at $T{=}0$ and 6.3% at $T{=}1$. The strong results on GSM8K suggest that GTO's draft-tree reward effectively captures sequential reasoning patterns critical for mathematical problem solving.

The results across diverse tasks and models highlight the versatility and robustness of GTO. The consistent improvements over the SoTA EAGLE-3, even at different temperatures, underscore GTO's effectiveness in handling varying levels of stochasticity in token predictions. Notably, the performance gains are more pronounced at temperature $T = 0$ across most settings, suggesting that GTO's deterministic tree optimization particularly benefits greedy decoding scenarios.

**Compatibility evaluation.** To further test compatibility and transferability, we evaluate GTO with draft models not initialized by EAGLE-3. Specifically, we fine-tune the draft models from two efficient speculative decoding methods—GRIFFIN and HASS—using GTO, and evaluate them under identical configurations on LLaMA-3-Instruct-8B and LLaMA-2-Chat-7B.

As shown in Table 2, both **GRIFFIN+GTO** and **HASS+GTO** achieve consistent gains over their baselines. At $T{=}0$, GRIFFIN+GTO improves the average speedup ratio ($SR$) and acceptance length ($\tau$) by 7.8% and 7.4%, respectively, while HASS+GTO improves them by 8.3% and 8.0%. At $T{=}1$, GRIFFIN+GTO increases $SR$ and $\tau$ by 5.0% and 5.6%, and HASS+GTO by 4.6% and 5.8%. These results validate GTO's compatibility and transferability across distinct draft backbones, establishing it as a general and effective approach for bridging the training-decoding tree-policy misalignment.

## 4.2 ABLATION STUDY

**Aggregation Operator.** We ablate the aggregation operator in the Draft Tree Reward (Sec. 3.1) on LLaMA-3.1-Instruct-8B. Our method employs the *smooth maximum* via log-sum-exp (LSE), which preserves differentiability while emphasizing strong branches (Eq. (5)). We compare against two alternatives under identical settings: (i) *Sum (Average)*: $\mathbf{r}_t^{\text{sum}} = \frac{1}{N}\sum_{i=1}^{N}\mathbf{L}_{t,i}$, treating all branches equally; (ii) *Max*: $\mathbf{r}_t^{\max} = \max_i \mathbf{L}_{t,i}$, focusing only on the best branch but non-smooth.

Across all benchmarks and decoding temperatures, LSE aggregation (GTO) attains the best speedup ratio ($SR$) and acceptance length ($\tau$). At $T{=}0$, GTO improves the average $SR$ by 2.1% over Max and 4.8% over Sum, with comparable gains in $\tau$. At $T{=}1$, the advantage remains, with $SR$ gains of 1.4% over Max and 3.8% over Sum, again accompanied by consistent improvements in $\tau$.

Table 4: Ablation of grouping size $m$ on LLaMA-3.1 8B.

| Method | Temperature = 0 | | | | | | | | Temperature = 1 | | | | | | | |
|---|---|---|---|---|---|---|---|---|---|---|---|---|---|---|---|---|
| | MT-bench | | HumanEval | | GSM8K | | Average | | MT-bench | | HumanEval | | GSM8K | | Average | |
| | $SR\uparrow$ | $\tau\uparrow$ | $SR\uparrow$ | $\tau\uparrow$ | $SR\uparrow$ | $\tau\uparrow$ | $SR\uparrow$ | $\tau\uparrow$ | $SR\uparrow$ | $\tau\uparrow$ | $SR\uparrow$ | $\tau\uparrow$ | $SR\uparrow$ | $\tau\uparrow$ | $SR\uparrow$ | $\tau\uparrow$ |
| $m=1$ | 3.32 | 5.94 | 4.02 | 6.71 | 3.47 | 6.25 | 3.60 | 6.30 | 2.40 | 4.54 | 3.06 | 5.71 | 2.91 | 5.55 | 2.79 | 5.27 |
| $m=4$ | 3.42 | 6.12 | 4.15 | 6.91 | 3.57 | 6.44 | 3.71 | 6.49 | 2.48 | 4.68 | 3.15 | 5.89 | 3.01 | 5.72 | 2.88 | 5.43 |
| $m=8$ (GTO) | **3.44** | **6.15** | **4.17** | **6.95** | **3.59** | **6.47** | **3.73** | **6.52** | **2.49** | **4.70** | **3.17** | **5.92** | **3.02** | **5.75** | **2.89** | **5.46** |
| $m=16$ | 3.27 | 5.84 | 3.96 | 6.60 | 3.41 | 6.15 | 3.55 | 6.20 | 2.37 | 4.47 | 3.01 | 5.62 | 2.87 | 5.46 | 2.75 | 5.18 |
| $m=32$ | 3.17 | 5.66 | 3.84 | 6.39 | 3.30 | 5.95 | 3.44 | 6.00 | 2.29 | 4.32 | 2.92 | 5.45 | 2.78 | 5.29 | 2.66 | 5.02 |

Table 5: Ablation of reward debiasing with a reference model on LLaMA-3.1 8B.

| Method | Temperature = 0 | | | | | | | | Temperature = 1 | | | | | | | |
|---|---|---|---|---|---|---|---|---|---|---|---|---|---|---|---|---|
| | MT-bench | | HumanEval | | GSM8K | | Average | | MT-bench | | HumanEval | | GSM8K | | Average | |
| | $SR\uparrow$ | $\tau\uparrow$ | $SR\uparrow$ | $\tau\uparrow$ | $SR\uparrow$ | $\tau\uparrow$ | $SR\uparrow$ | $\tau\uparrow$ | $SR\uparrow$ | $\tau\uparrow$ | $SR\uparrow$ | $\tau\uparrow$ | $SR\uparrow$ | $\tau\uparrow$ | $SR\uparrow$ | $\tau\uparrow$ |
| **GTO (Debiased)** | **3.44** | **6.15** | **4.17** | **6.95** | **3.59** | **6.47** | **3.73** | **6.52** | **2.49** | **4.70** | **3.17** | **5.92** | **3.02** | **5.75** | **2.89** | **5.46** |
| w/o Debiasing | 3.30 | 5.78 | 3.84 | 6.62 | 3.50 | 6.23 | 3.55 | 6.21 | 2.39 | 4.53 | 3.03 | 5.64 | 2.87 | 5.35 | 2.76 | 5.17 |

These results highlight the trade-offs of alternative operators: *Sum* dilutes signal by averaging weak branches, while *Max* is brittle and non-smooth, overfitting to a single path with poor gradient coverage. In contrast, LSE interpolates between them, providing a stable and selective objective that better aligns with decoding-time re-ranking and pruning.

**Group Size.** We ablate the group size $m$ in Tree Reward Optimization (Sec. 3.2) on LLaMA-3.1-Instruct 8B with $m \in \{1, 4, 8, 16, 32\}$. As shown in Table 4, the default $m=8$ of GTO achieves the best average $SR$ and $\tau$, while $m=4$ is within $< 1\%$, indicating a stable plateau. In contrast, $m=1$ and $m=16$ show clear degradation, and $m=32$ performs worst.

Small groups (e.g., $m=1$) suffer from noisy, context-misaligned rewards, weakening credit assignment. Large groups (e.g., $m \geq 16$) span longer contexts, introducing drift and bias that hurt learning. Thus, moderate sizes ($m \in [4, 8]$) strike the best balance between variance reduction and context alignment, yielding the most reliable gains in $SR$ and $\tau$.

These observations align with the theoretical insights from the GRPO (Shao et al., 2024), which shows that very small groups suffer from high-variance and unstable updates, while very large groups suffer from signal attenuation and slower convergence due to excessive averaging. In our GTO experiments, we also observe this qualitative pattern. Group sizes in the range of 4–8 consistently provide the best balance: they significantly reduce variance relative to size 1, yet still preserve enough reward contrast to produce strong learning signals. Larger groups (e.g., size 32) remain stable but deliver noticeably weaker improvements due to the normalization-induced compression described above.

**Reward Debiasing.** We ablate the reward shaping and standardization step (Eq. (9)) in Tree Reward Optimization on LLaMA-3.1-Instruct-8B. Debiasing computes a control-variated reward by subtracting the tree-level reward of a frozen reference draft model $\mathcal{M}_0$ (Phase I) from the current model $\mathcal{M}$ for matched prefixes, reducing variance and improving credit assignment. We compare our default GTO (Debiased) against a variant that omits this subtraction (w/o Debiasing), with all other settings fixed.

As shown in Table 5, debiasing consistently improves both $SR$ and $\tau$. At $T=0$, GTO achieves $+5.0\%$ $SR$ and $+5.1\%$ $\tau$ over w/o Debiasing; at $T=1$, the gains are $+5.6\%$ and $+4.7\%$. Without debiasing, rewards are noisier and context-dependent, yielding weaker draft policies and shorter acceptance lengths.

**Continual training.** To ensure that the performance gains of GTO stem from our proposed algorithmic improvements rather than the additional computational budget, we introduce a continual training baseline. Specifically, we further fine-tune the vanilla EAGLE-3 draft model for an additional 200 A100-80G GPU hours. This baseline employs the exact same training data and strictly follows the original EAGLE-3 training recipe for the DeepSeek-R1-Distill-LLaMA-8B target model.

Table 6: Ablation of continual training on draft model.

| Method | Temperature = 0 | | | | | | | | Temperature = 1 | | | | | | | |
|---|---|---|---|---|---|---|---|---|---|---|---|---|---|---|---|---|
| | MT-bench | | HumanEval | | GSM8K | | Average | | MT-bench | | HumanEval | | GSM8K | | Average | |
| | $SR\uparrow$ | $\tau\uparrow$ | $SR\uparrow$ | $\tau\uparrow$ | $SR\uparrow$ | $\tau\uparrow$ | $SR\uparrow$ | $\tau\uparrow$ | $SR\uparrow$ | $\tau\uparrow$ | $SR\uparrow$ | $\tau\uparrow$ | $SR\uparrow$ | $\tau\uparrow$ | $SR\uparrow$ | $\tau\uparrow$ |
| **GTO** | **3.49** | **5.60** | **3.98** | **6.58** | **4.20** | **6.92** | **3.89** | **6.37** | **2.76** | **4.59** | **3.34** | **5.44** | **3.71** | **6.50** | **3.27** | **5.51** |
| EAGLE-3 | 3.34 | 5.32 | 3.59 | 5.88 | 3.78 | 6.16 | 3.57 | 5.79 | 2.71 | 4.54 | 3.15 | 5.10 | 3.49 | 5.82 | 3.11 | 5.15 |
| continual training | 3.37 | 5.38 | 3.61 | 5.94 | 3.81 | 6.24 | 3.60 | 5.85 | 2.70 | 4.53 | 3.16 | 5.15 | 3.54 | 5.93 | 3.13 | 5.20 |

As reported in Table 6, GTO achieves a consistently higher speedup (approximately 4%–7% on average) across various benchmarks and temperatures compared to this continued-training baseline. Furthermore, we observe that the continual training baseline performs almost identically to the vanilla EAGLE-3. This behavior is expected: because the training dataset is already encompassed within EAGLE-3's original training mixture, the draft model is already near convergence on this distribution. Without the introduction of novel data, allocating additional compute to the standard training objective merely reinforces an already-optimized model, yielding negligible improvements.

These findings confirm that the efficacy of GTO does not derive from simply extending training time or data exposure. Instead, the improvements are fundamentally algorithmic, rooted in GTO's ability to explicitly align the draft tree policy with the target acceptance dynamics, thereby directly resolving the policy misalignment that typically bottlenecks speculative decoding.

## 5 CONCLUSION

In this paper, we proposed **Group Tree Optimization** (GTO) to bridge the draft policy misalignment between training and decoding. GTO introduces a decoding-faithful *Draft Tree Reward* that directly optimizes the expected acceptance length and a stable *group-based optimization* that contrasts current and reference trees, standardizes advantages across nearby contexts, and updates via a PPO-style clipped surrogate along the longest accepted sequence. Extensive evaluations across diverse LLMs and datasets show that GTO consistently outperforms SoTAs, achieving the highest speedup ratios and acceptance lengths.

**Limitations.** GTO increases training-time compute due to its two-phase procedure and the need to construct and evaluate grouped draft trees during training. Nevertheless, GTO is *model-agnostic* and complementary to existing speculative decoding methods: it can be directly fine-tuned on top of pretrained draft models (e.g., EAGLE-3, GRIFFIN) without architectural changes or modifications to the verification stack. In practice, the draft model is trained once, whereas decoding dominates the runtime in real-world deployments; the added training cost is therefore amortized by improved inference efficiency. In our experiments, GTO improves the speedup ratio by more than 7% over EAGLE-3, making the extra training cost a reasonable trade-off for latency-sensitive applications.

## ACKNOWLEDGEMENT

This work was supported by the Yangtze River Delta Science and Technology Innovation Community Joint Research Project (YDZX20233100004031) and the Singapore Ministry of Education (MOE) Academic Research Fund (AcRF) Tier 1 grant (Proposal ID: 25-SIS-SMU-003). Any opinions, findings and conclusions or recommendations expressed in this material are those of the author(s) and do not reflect the views of the Ministry of Education, Singapore.

## ETHICS STATEMENT

GTO improves *efficiency* of large language model decoding. Nevertheless, faster generation could increase the throughput of undesirable content if deployed without safeguards. We recommend deploying GTO only with established safety measures (content filters, rate limiting, audit logging, and red-teaming) and within the original safety and usage policies of the underlying models.

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

## A    PROOF OF THEOREM 1

We first make explicit the objects in play. Let the draft tree at step $t$ have $N$ branches (root-to-leaf paths) indexed by $i \in [N]$. For each branch $i$, let $\mathbf{z}_{i,1:\ell_i}$ denote its token sequence up to depth $\ell_i$, and let

$$\mathbf{L}_{t,i} \in \{0, 1, \ldots, d\}$$

denote the (random or deterministic) number of consecutive tokens, starting at the current prefix, that the target model would accept if branch $i$ were proposed. The draft-tree reward is the smooth maximum

$$\mathbf{r}_t = \frac{1}{\eta} \log\Big( \sum_{i=1}^{N} e^{\eta \mathbf{L}_{t,i}} \Big) \quad \text{with} \quad \eta > 0,$$

which satisfies the standard bounds

$$\max_i \mathbf{L}_{t,i} \leq \mathbf{r}_t \leq \max_i \mathbf{L}_{t,i} + \frac{1}{\eta} \log N. \tag{1}$$

For decoding, define for each $j \geq 1$ the event

$$\mathcal{E}_j(\mathbf{T}_t) = \{\text{at least } j \text{ tokens are accepted at decoding}\}.$$

Then the expected acceptance length under target temperature $T$ can be expressed as

$$\mathbb{E}\big[ L_T^{\text{dec}}(\mathbf{T}_t) \big] = \sum_{j=1}^{d} \mathbb{P}_T(\mathcal{E}_j(\mathbf{T}_t)). \tag{2}$$

We will use the following elementary monotonicity fact.

**Lemma 1** (Coordinate-wise monotonicity of acceptance probability). *Fix a draft tree topology and branch token sequences $\{\mathbf{z}_{i,1:\ell_i}\}_{i=1}^{N}$. For any $j \geq 1$, the event $\mathcal{E}_j(\mathbf{T}_t)$ can be written as the union*

$$\mathcal{E}_j(\mathbf{T}_t) = \bigcup_{i=1}^{N} \mathcal{B}_{i,j}, \qquad \mathcal{B}_{i,j} := \{\text{the target rollout matches } \mathbf{z}_{i,1:j}\}.$$

*If we increase a single coordinate $\mathbf{L}_{t,i}$ by $\Delta \in \mathbb{N}$ (keeping other $\mathbf{L}_{t,k}$ fixed), then for each $j \in \{\mathbf{L}_{t,i}+1, \ldots, \mathbf{L}_{t,i}+\Delta\}$, the union gains a new set $\mathcal{B}_{i,j}$ and hence*

$$\mathbb{P}_T(\mathcal{E}_j(\mathbf{T}_t)) \text{ is non-decreasing.}$$

*Moreover, if $T > 0$ (softmax sampling with strictly positive support over tokens), then $\mathbb{P}_T(\mathcal{B}_{i,j}) > 0$ and thus $\mathbb{P}_T(\mathcal{E}_j(\mathbf{T}_t))$ increases strictly for those $j$.*

*Proof sketch.* For each $i$, the event $\mathcal{B}_{i,j}$ corresponds to the target producing the specific $j$-token prefix $\mathbf{z}_{i,1:j}$. Increasing $\mathbf{L}_{t,i}$ by $\Delta$ adds new prefixes at depths $\mathbf{L}_{t,i}+1, \ldots, \mathbf{L}_{t,i}+\Delta$, hence enlarging the union. Under $T > 0$, each concrete token sequence has strictly positive probability under a softmax LM, so the probability mass added is positive. Disjointness at the level of exact token sequences follows from the tree structure: no two distinct branches share the same length-$j$ token prefix, so $\mathcal{B}_{i,j}$ is not a subset of $\bigcup_{k \neq i} \mathcal{B}_{k,j}$. $\square$

We now prove the two cases in Theorem 1.

*Proof of Theorem 1.* (a) $T > 0$. The function $\mathbf{r}_t = \frac{1}{\eta} \log\big( \sum_i e^{\eta \mathbf{L}_{t,i}} \big)$ is strictly increasing in each coordinate $\mathbf{L}_{t,i}$. Because $\mathbf{L}_{t,i}$ are integer-valued lengths, any increase in $\mathbf{r}_t$ implies that at least one coordinate $\mathbf{L}_{t,i}$ increases by an integer $\Delta \geq 1$.[1] By Lemma 1, for each newly covered depth $j \in \{\mathbf{L}_{t,i}+1, \ldots, \mathbf{L}_{t,i}+\Delta\}$ we have $\mathbb{P}_T(\mathcal{E}_j(\mathbf{T}_t))$ increases strictly (because $T > 0$ confers strictly positive mass on the corresponding prefix event). Summing these strictly positive increases over

---

[1] Formally, along any path that increases $\mathbf{r}_t$, the first time $\mathbf{r}_t$ changes must coincide with an increment in at least one discrete coordinate.

$j$ and possibly over multiple improved branches (if several coordinates increased) and invoking equation 2 yields

$$\mathbb{E}\big[L_T^{\mathrm{dec}}(\mathbf{T}_t)\big] \quad \text{increases strictly whenever } \mathbf{r}_t \text{ increases.}$$

(b) $T = 0$. Let $\mathbf{s}^\star$ be the unique greedy target trajectory. Then $\mathbf{L}_{t,i}$ equals the longest common-prefix length between branch $i$ and $\mathbf{s}^\star$, and

$$\mathbb{E}\big[L_0^{\mathrm{dec}}(\mathbf{T}_t)\big] \;=\; \max_i \mathbf{L}_{t,i}.$$

Using the smooth-max bounds equation 1 with $M := \max_i \mathbf{L}_{t,i}$, we have

$$M \;\leq\; \mathbf{r}_t \;\leq\; M + \frac{1}{\eta}\log N.$$

Consequently, if $\mathbf{r}_t$ increases by more than the residual slack-to-plateau,

$$\Delta\mathbf{r}_t \;>\; \Big(M + \frac{1}{\eta}\log N\Big) - \mathbf{r}_t,$$

then the new reward $\mathbf{r}'_t$ must satisfy $\mathbf{r}'_t > M + \frac{1}{\eta}\log N$, which is impossible unless the new maximum increases to $M' \geq M + 1$. Hence, under $T = 0$,

$$\mathbf{r}'_t - \mathbf{r}_t \;>\; \Big(M + \tfrac{1}{\eta}\log N\Big) - \mathbf{r}_t \;\implies\; \mathbb{E}\big[L_0^{\mathrm{dec}}(\mathbf{T}_t)\big] = \max_i \mathbf{L}_{t,i} \text{ strictly increases.}$$

This gives a simple sufficient condition: an increase in $\mathbf{r}_t$ that exceeds the softmax slack $\frac{1}{\eta}\log N - (\mathbf{r}_t - M)$ necessarily raises the deterministic acceptance length.

Putting (a) and (b) together, we obtain the stated guarantees: for $T > 0$, any increase in $\mathbf{r}_t$ strictly increases the expected acceptance length; for $T = 0$, an increase in $\mathbf{r}_t$ that exceeds the smooth-max slack forces an increase in $\max_i \mathbf{L}_{t,i}$. $\qquad\square$

**Remarks.** (i) The case $T > 0$ relies only on the strictly positive support of the target sampler; it holds for any softmax temperature $T > 0$ (or any sampler with full support). (ii) The sufficient condition in $T = 0$ is tight with respect to the standard smooth-max bounds equation 1; no stronger implication can be made from $\mathbf{r}_t$ alone because $\mathbf{r}_t$ can increase by raising only sub-maximal branches without changing the maximum.

## B  IMPLEMENTATION DETAIL

### B.1  DRAFT TREE STRUCTURE

Across all experiments, we adopt a dynamic draft tree with a fixed budget of 60 draft tokens, a maximum tree depth of 7 and top-$k$ of 10, following the configuration shown to be effective in EAGLE-3.

### B.2  TOKEN-LEVEL LOSS IN EQ. (13)

Let $\mathcal{D}$ be the training corpus over a vocabulary $\mathcal{V}$. For a sequence $\mathbf{x} = (x_1, \ldots, x_L) \in \mathcal{D}$, denote the prefix $\mathbf{x}_{1:i-1} = (x_1, \ldots, x_{i-1})$. Let $p_{\mathcal{T}}(\cdot \mid \mathbf{x}_{1:i-1})$ and $p_{\mathcal{M}}(\cdot \mid \mathbf{x}_{1:i-1})$ be the next-token distributions produced by the target model $\mathcal{T}$ and the draft model $\mathcal{M}$, respectively, under the *same* teacher-forced prefix. We define the token-level loss as the expected cross-entropy from the teacher to the student:

$$\mathcal{L}_{\mathrm{token}} \;=\; \mathbb{E}_{\mathbf{x} \sim \mathcal{D}}\left[\frac{1}{|\mathcal{I}(\mathbf{x})|}\sum_{i \in \mathcal{I}(\mathbf{x})} H\big(p_{\mathcal{T}}(\cdot \mid \mathbf{x}_{1:i-1}), p_{\mathcal{M}}(\cdot \mid \mathbf{x}_{1:i-1})\big)\right],$$

where $\mathcal{I}(\mathbf{x}) \subseteq \{1, \ldots, L\}$ indexes supervised positions (e.g., all non-padding positions) and

$$H(p, q) \;=\; -\sum_{v \in \mathcal{V}} p(v)\log q(v)$$

is the cross-entropy. Equivalently, since $H(p_\mathcal{T}, p_\mathcal{M}) = \mathrm{KL}(p_\mathcal{T} \| p_\mathcal{M}) + H(p_\mathcal{T})$ and $H(p_\mathcal{T})$ does not depend on $\mathcal{M}$, minimizing $\mathcal{L}_{\mathrm{token}}$ is equivalent (up to an additive constant) to minimizing

$$\mathbb{E}_{\mathbf{x} \sim \mathcal{D}} \left[ \frac{1}{|\mathcal{I}(\mathbf{x})|} \sum_{i \in \mathcal{I}(\mathbf{x})} \mathrm{KL}(p_\mathcal{T}(\cdot \mid \mathbf{x}_{1:i-1}) \| p_\mathcal{M}(\cdot \mid \mathbf{x}_{1:i-1})) \right].$$

## B.3 TRAINING CONFIGURATION

We fine-tune the draft model with AdamW and a warmup–decay schedule under mixed precision and ZeRO optimizations. Key hyperparameters are summarized below:

- Draft-tree construction: top-$k$ for per-node expansion set to $k = 10$.
- Draft-tree reranking: top-$g$ candidates per step set to $g = 60$.
- Smooth-max temperature in tree reward: $\eta = 1$.
- Number of groups per sequence: $K = 16$.
- Group size (prefixes per group): $m = 8$.
- Scalar weight on the GTO loss: $\omega = 0.5$ in $\mathcal{L} = \mathcal{L}_{\mathrm{token}} + \omega\,\mathcal{L}_{\mathrm{GTO}}$.

**Optimizer and scheduler.**

- Optimizer: AdamW with $\beta_1{=}0.9$, $\beta_2{=}0.95$, weight decay $= 0$.
- Learning rate: Warm up linearly from $0$ to $5{\times}10^{-6}$ over 1,000 steps, then decay over a total of 60,000 steps.
- Gradient clipping: $0.5$.

**Precision and parallelism.**

- Mixed precision: FP16 autocast with dynamic loss scaling (initial scale $2^{14}$; window $=$ 1000; hysteresis $= 2$; min scale $= 1$).
- ZeRO: Stage-2 with overlapping communication, all-gather/reduce-scatter enabled; bucket sizes $2{\times}10^8$.
- Gradient accumulation: 2 steps; per-GPU micro-batch size: 1.

**Training loop.**

- Epochs: 5
- max sequence length: 2048
- dataloader workers: 2

Additional hyperparameters and scripts are available at https://github.com/hsj576/GTO.

The full GTO update is summarized in Algorithm 1.

## C CLARIFICATION OF BASELINE METHODS

For EAGLE, EAGLE-2, EAGLE-3, HASS, GRIFFIN, Medusa and Hydra, we directly utilized the publicly released draft model parameters provided by the respective authors. For methods that do not require draft model training, such as PLD, Lookahead, and SPS, we evaluated performance using official code from their GitHub repositories.

---

**Algorithm 1** GTO Phase II: Group-based Optimization of Draft Tree Reward

---

**Require:** Draft model $\mathcal{M}$, reference draft model $\mathcal{M}_0$, target model $\mathcal{T}$, group size $m$, clip $\epsilon$, std floor $\delta$, reward aggregator $\mathcal{R}$
 1: **for** each minibatch of training sequences **do**
 2:     **for** each sequence $\mathbf{x}$ in batch **do**
 3:         Sample $\{\mathbf{G}^{(k)}\}_{k=1}^{K} \leftarrow \text{SampleGroups}(\mathbf{x}, m)$         $\triangleright \mathbf{G}^{(k)} = \{t_k, \ldots, t_k + m - 1\}$
 4:         **for** each group $\mathbf{G}^{(k)}$ **do**
 5:             **for** each $i \in \mathbf{G}^{(k)}$ **do**
 6:                 Build trees: $\mathbf{T}_i \leftarrow \mathcal{G}(\mathcal{M}, \mathbf{x}_{1:i})$, $\bar{\mathbf{T}}_i \leftarrow \mathcal{G}(\mathcal{M}_0, \mathbf{x}_{1:i})$
 7:                 Compute rewards: $\mathbf{r}_i \leftarrow \mathcal{R}(\mathbf{T}_i)$, $\bar{\mathbf{r}}_i \leftarrow \mathcal{R}(\bar{\mathbf{T}}_i)$
 8:                 Debiased reward: $\mathbf{R}_i \leftarrow \mathbf{r}_i - \bar{\mathbf{r}}_i$
 9:                 Find longest accepted sequence $\widehat{\mathbf{S}}_i$ in $\mathbf{T}_i$ and its length $l_i$
10:                 Likelihood ratio: $s_i \leftarrow \exp\big((\log \mathcal{M}(\widehat{\mathbf{S}}_i | \mathbf{x}_{1:i}) - \log \mathcal{M}_0(\widehat{\mathbf{S}}_i | \mathbf{x}_{1:i}))/l_i\big)$
11:             **end for**
12:             Standardize within group: $\mathcal{A}_i \leftarrow \big(\mathbf{R}_i - \text{mean}(\{\mathbf{R}_j\})\big)/\big(\text{std}(\{\mathbf{R}_j\}) + \delta\big)$
13:             Compute group loss: $\mathcal{L}_{\text{GTO}} \leftarrow -\frac{1}{m}\sum_{i \in \mathbf{G}^{(k)}} \min\big(s_i \mathcal{A}_i, \text{clip}(s_i, 1 - \epsilon, 1 + \epsilon)\mathcal{A}_i\big)$
14:         **end for**
15:     **end for**
16:     Update $\mathcal{M}$ by minimizing $\mathcal{L} = \mathcal{L}_{\text{token}} + \omega \mathcal{L}_{\text{GTO}}$
17: **end for**

---

# D  TRAINING OVERHEAD OF GTO

**Compute budget.** All results were obtained on NVIDIA A100 80 GB GPUs under mixed precision with ZeRO-2. The Phase-II GTO fine-tuning requires approximately *(i)* 200 GPU-hours for 8B models, *(ii)* 400 GPU-hours for 13B models, and *(iii)* 900 GPU-hours for 70B models. These compute budgets cover end-to-end GTO training (including grouped tree construction and verification) and exclude any pretraining of the base or drafter models, as we fine-tune on publicly available pretrained drafters.

**Why the overhead is worthwhile.**

- **Model-agnostic and complementary.** *GTO is model-agnostic and complementary to existing speculative decoding methods*: it can be directly fine-tuned on top of pretrained draft models (e.g., EAGLE-3, GRIFFIN) **without architectural changes** or modifications to the verification stack.

- **Amortized cost in deployment.** *Train once, use everywhere*: the draft model is trained a single time, whereas decoding dominates the runtime in real-world deployments; the added training cost is therefore **amortized by improved inference efficiency**.

- **Measured gains.** In our experiments, GTO delivers $> \mathbf{7}\%$ higher end-to-end speedup ratio than EAGLE-3, making the small additional training budget a **favorable trade-off** for latency-sensitive applications.

# E  ABLATION ON TREE CONFIGURATION

In our primary experiments, we default to the standard EAGLE tree configuration (e.g., depth 7, 60 total tokens). This tree structure has emerged as the de facto standard in modern speculative decoding research and is widely adopted by recent state-of-the-art frameworks (e.g., EAGLE-3, HASS, GRIFFIN). By deliberately adopting this configuration, we ensure that our results are strictly comparable to established baselines, accurately reflect realistic deployment settings, and maximize practical relevance for the community.

**Tree-Agnostic Formulation.** Importantly, GTO is not inherently constrained to any specific tree layout. Methodologically, GTO optimizes the drafter to maximize the expected acceptance length

Table 7: Ablation of Tree Configuration $m$ on LLaMA-3.1 8B.

| Depth | Total Tokens | Temperature = 0 | | | | | | | | Temperature = 1 | | | | | | | |
|---|---|---|---|---|---|---|---|---|---|---|---|---|---|---|---|---|---|
| | | MT-bench | | HumanEval | | GSM8K | | Average | | MT-bench | | HumanEval | | GSM8K | | Average | |
| | | $SR\uparrow$ | $\tau\uparrow$ | $SR\uparrow$ | $\tau\uparrow$ | $SR\uparrow$ | $\tau\uparrow$ | $SR\uparrow$ | $\tau\uparrow$ | $SR\uparrow$ | $\tau\uparrow$ | $SR\uparrow$ | $\tau\uparrow$ | $SR\uparrow$ | $\tau\uparrow$ | $SR\uparrow$ | $\tau\uparrow$ |
| 6 | 50 | 3.29 | 5.81 | 3.71 | 6.45 | 3.46 | 6.12 | 3.49 | 6.13 | 2.39 | 4.54 | 3.10 | 5.75 | 2.90 | 5.40 | 2.80 | 5.23 |
| 6 | 60 | 3.32 | 5.86 | 3.73 | 6.50 | 3.49 | 6.21 | 3.51 | 6.19 | 2.41 | 4.56 | 3.12 | 5.79 | 2.93 | 5.51 | 2.82 | 5.29 |
| 6 | 70 | 3.34 | 5.93 | 3.73 | 6.52 | 3.51 | 6.27 | 3.53 | 6.24 | 2.41 | 4.56 | 3.12 | 5.80 | 2.93 | 5.52 | 2.82 | 5.29 |
| 7 | 50 | 3.39 | 6.05 | 3.95 | 6.87 | 3.56 | 6.39 | 3.63 | 6.44 | 2.44 | 4.60 | 3.15 | 5.86 | 2.83 | 5.30 | 2.80 | 5.25 |
| 7 | 60 | 3.44 | 6.15 | 4.17 | 6.95 | 3.59 | 6.47 | 3.73 | 6.52 | 2.49 | 4.70 | 3.17 | 5.92 | 3.02 | 5.75 | 2.89 | 5.46 |
| 7 | 70 | 3.46 | 6.23 | 4.17 | 7.00 | 3.63 | 6.57 | 3.76 | 6.60 | 2.51 | 4.74 | 3.24 | 6.10 | 3.10 | 5.86 | 2.95 | 5.57 |
| 8 | 50 | 3.56 | 6.22 | 4.22 | 7.24 | 3.61 | 6.59 | 3.80 | 6.68 | 2.49 | 4.72 | 3.29 | 6.34 | 2.80 | 5.24 | 2.86 | 5.43 |
| 8 | 60 | 3.54 | 6.34 | 4.24 | 7.33 | 3.66 | 6.69 | 3.81 | 6.79 | 2.46 | 4.73 | 3.20 | 6.12 | 2.95 | 5.56 | 2.87 | 5.47 |
| 8 | 70 | 3.49 | 6.43 | 4.20 | 7.39 | 3.71 | 6.83 | 3.80 | 6.88 | 2.46 | 4.74 | 3.22 | 6.28 | 2.78 | 5.26 | 2.82 | 5.43 |

under speculative decoding. Because the training procedure learns underlying acceptance-related to-ken distribution patterns rather than memorizing a fixed tree shape, GTO is conceptually compatible with any tree configuration that satisfies two basic conditions: (i) the drafter can successfully con-struct the tree, and (ii) a tree-level reward can be computed via target model verification. This design ensures that GTO's optimization principles generalize well beyond the default EAGLE layout.

**Interpolation and Extrapolation Performance.** To empirically validate GTO's robustness to varying tree structures, we conduct an ablation study exploring the extrapolation and interpolation of the tree configuration at inference time. Specifically, we take a drafter trained exclusively on the standard EAGLE tree (depth 7, 60 total tokens) and evaluate it using modified configurations during inference. We vary the tree depth $\in \{6, 7, 8\}$ and the total drafted tokens per step $\in \{50, 60, 70\}$. The comprehensive results across different benchmarks and temperatures are presented in Table 7. We observe four key findings:

- **Consistent Performance:** GTO maintains strong speedup ($SR$) and acceptance rate ($\tau$) improvements across all 9 tested configurations, demonstrating high robustness to both depth and width variations.

- **Graceful Interpolation:** Configurations closely interpolating the training setting exhibit minimal performance degradation, with speedup varying by only $1\%$–$3\%$ from the optimal setting.

- **Effective Extrapolation:** When extrapolating to depth 8 (beyond the training depth of 7), GTO still achieves competitive or even superior performance. This indicates that the token-level alignment patterns learned by GTO transfer effectively to deeper and larger trees.

- **Stability Across Conditions:** The robustness holds consistently across MT-bench, Hu-manEval, and GSM8K, as well as under both greedy ($T = 0$) and sampling ($T > 0$) de-coding settings. This confirms that the generalization is neither task-specific nor decoding-strategy-specific.

In conclusion, these results demonstrate that GTO does not overfit to a specific tree configuration during the training phase. The learned alignment strategies remain highly effective when the tree structure is modified at inference time. Consequently, GTO fully supports dynamic tree config-uration updates during inference while maintaining, and in some cases even improving, overall speculative decoding performance.

## F  LLM USAGE STATEMENT

Large language models were used minimally for proofreading and grammar checking. The research ideas, methodology, experiments, and analysis were entirely conceived and conducted by the au-thors.

