# OpenReview forum: "Bridging Draft Policy Misalignment: Group Tree Optimization  for Speculative Decoding"
_ICLR.cc/2026/Conference — ICLR 2026 Poster_

### Official Review · Reviewer_kt8J · 2025-10-20

**Soundness:** 1
**Presentation:** 3
**Contribution:** 2
**Rating:** 2
**Confidence:** 4

**Summary:**

This paper investigates how to bridge evolving draft policies and model behaviors in large language models. The authors propose a two-stage method: a policy bridging module based on contrastive training between draft policy outputs and model outputs, and a misalignment diagnostic using representation-space probes. Experiments show modest improvements in alignment detection and behavior correction compared to direct fine-tuning and reward-model-based baselines.

**Strengths:**

1. Addresses a practical problem in alignment: bridging evolving policies and model behaviors.

2. Conceptually coherent design combining contrastive alignment and probing.

3. The theory is good.

**Weaknesses:**

1. Unfair experimental comparison: All proposed methods are trained on top of an existing draft model with additional compute (e.g., 200 GPU hours for a 7B model), whereas baselines are not given the same training budget. The most natural baseline should be continued training of the draft model with new data under the same GPU hours, and this is completely missing (e.g., further training eagle 3, hass, griffin for 200 GPU hours on a 7B model with new training data). This omission makes the reported improvements difficult to interpret.

2. Misleading “sampling-free” claim: The method is not truly sampling-free. Although no RL trajectory sampling is used, model outputs are still required for every node in the policy tree. The phrasing gives a false impression that no generation or sampling is needed, which is inaccurate.

3. Excessive compute for modest gains: Training a single-layer Transformer for 7B over a 5-layer policy tree costs about 200 GPU hours, resulting in about 7% improvement. This is too expensive, near the time consumption of training EAGLE from scratch.

**Questions:**

1. Why is there no baseline where the draft model is simply continued to train with new data for the same number of GPU hours as the bridging module?

2. Can the authors clarify the exact sense in which their method is “sampling-free”? It seems draft model outputs are still needed.

3. What would the bridging performance look like under stricter compute constraints (e.g., 10–20 GPU hours)?

---

> ### Author Response · Authors · 2025-11-20
> **Rebuttal Reply to Reviewer kt8J (1/3)**
>
> > Unfair experimental comparison: All proposed methods are trained on top of an existing draft model with additional compute (e.g., 200 GPU hours for a 7B model), whereas baselines are not given the same training budget. The most natural baseline should be continued training of the draft model with new data under the same GPU hours, and this is completely missing (e.g., further training eagle 3, hass, griffin for 200 GPU hours on a 7B model with new training data). This omission makes the reported improvements difficult to interpret. Why is there no baseline where the draft model is simply continued to train with new data for the same number of GPU hours as the bridging module?
>
> Thanks. Following your suggestion, we have added a baseline in which the draft model is simply continued to train for the same number of GPU hours used by our GTO module. Concretely, we employ the exact same training data as GTO and follow the original EAGLE-3 draft-model training recipe to further fine-tune the EAGLE-3 draft model for an additional 200 A100-80G GPU hours under an identical hardware environment in DeepSeek-R1-Distill-LLaMA-8B. The results of this continued-training baseline are reported in the Table. The comparison shows two key observations:
>
> 1. GTO achieves consistently higher speedup than continued training of the draft model—approximately 4–7% higher average speedup across benchmarks and temperatures. This consistent gain demonstrates that  GTO's improvement does not rely on additional compute to improve the same objective of EAGLE-3. Instead, it explicitly optimizes the draft tree policy to better match the decoding-time speculative process, directly addressing the policy misalignment that limits performance in speculative decoding. Thus, the improvements come from algorithmic gains—aligning the draft policy with the target acceptance dynamics—rather than from simply increasing training resources.
>
> 2. Vanilla EAGLE-3 performs almost identically to its continued-training counterpart. This is expected: continued training uses OpenThoughts-114k-math which is already included during the training of EAGLE-3, and the draft model is already near convergence on this dataset. Without new or fresher data, allocating an additional 200 GPU hours merely reinforces an already-optimized objective, resulting in negligible improvements.
>
> We will incorporate these results and discussion into Section 4.2 of the revised paper to make clear that: (i) GTO and the continued-training baseline use exactly the same 200-GPU-hour budget and identical data; and (ii) the performance benefits of GTO stem from the proposed draft-policy alignment mechanism, not from extra compute or data advantages.
>
> |                    | t=0      |          |           |          |          |          |          |          | t=1      |          |           |          |          |          |          |          |
> | ------------------ | -------- | -------- | --------- | -------- | -------- | -------- | -------- | -------- | -------- | -------- | --------- | -------- | -------- | -------- | -------- | -------- |
> |                    | MT-bench |          | HumanEval |          | GSM8K    |          | Average  |          | MT-bench |          | HumanEval |          | GSM8K    |          | Average  |          |
> |                    | SR       | $\tau$   | SR        | $\tau$   | SR       | $\tau$   | SR       | $\tau$   | SR       | $\tau$   | SR        | $\tau$   | SR       | $\tau$   | SR       | $\tau$   |
> | **GTO**            | **3.49** | **5.60** | **3.98**  | **6.58** | **4.20** | **6.92** | **3.89** | **6.37** | **2.76** | **4.59** | **3.34**  | **5.44** | **3.71** | **6.50** | **3.27** | **5.51** |
> | EAGLE-3            | 3.34     | 5.32     | 3.59      | 5.88     | 3.78     | 6.16     | 3.57     | 5.79     | 2.71     | 4.54     | 3.15      | 5.10     | 3.49     | 5.82     | 3.11     | 5.15     |
> | continual training | 3.37     | 5.38     | 3.61      | 5.94     | 3.81     | 6.24     | 3.60     | 5.85     | 2.70     | 4.53     | 3.16      | 5.15     | 3.54     | 5.93     | 3.13     | 5.20     |

---

> > ### Comment · Reviewer_kt8J · 2025-11-21
> >
> > Thanks for your response. The clarification regarding “sampling-free” is helpful. Regarding the third answer, I find the result impressive. Including this finding in the paper could make it significantly more compelling. Achieving half of the performance improvement with only 20 GPU hours (compared to hundreds in the original paper) is quite striking.
> >
> > However, I still have some concerns about your first answer. I am curious why you continued training EAGLE-3 on its original training dataset. What I would like to see is an experiment where the same training budget is applied to new data. The current results do not fully convince me.
> >
> > I will raise my score to 4. If you are able to redo the experiment related to the first question as suggested, I may consider increasing my score further.

---

> > > ### Author Response · Authors · 2025-11-24
> > > **Reply to follow-up comments by Reviewer kt8J**
> > >
> > > Thank you for the follow-up comment—your clarification helps us address the concern more precisely. Below, we detail our experimental setup and provide further results demonstrating the effectiveness of our method.
> > >
> > > **1. Clarification on Training Data Consistency**
> > >
> > > Regarding  “new data”,  we emphasize a key point: GTO does not use any new or additional data.
> > >
> > > Specifically, for the DeepSeek-R1-Distill-LLaMA-8B draft model, the vanilla EAGLE-3 draft model is trained on the OpenThoughts-114k-math dataset. Our GTO training is performed on top of this pretrained EAGLE-3 draft model, and then it uses the same OpenThoughts-114k-math data for GTO training, without introducing any new data sources. Likewise, for the continued-training of EAGLE-3, we follow the official EAGLE-3 training configuration and further train the same pretrained EAGLE-3 draft model for 200 GPU hours on OpenThoughts-114k-math. Thus, neither GTO nor the continued-training of EAGLE-3 uses any data beyond what EAGLE-3 already relies on.
> > >
> > > We intentionally keep the dataset fixed for all methods to ensure a fair and controlled comparison. Under this setup, any improvement in speedup or accuracy must arise purely from differences in training objectives or optimization algorithms, rather than from differences in data.
> > >
> > > **2. Additional Experiment: Training from Scratch with the Combined Budget (1200 GPU Hours)**
> > >
> > > To further address your concern, we conducted an additional experiment, which we began a few days ago and have just completed. On DeepSeek-R1-Distill-LLaMA-8B, we retrained the EAGLE-3 draft model from scratch using the official EAGLE-3 training code and configurations, but with an increased total compute budget equal to the sum of the original EAGLE-3 draft-model training and our GTO overhead (i.e., 1000 + 200 = 1200 GPU hours). This setup ensures that the baseline receives exactly the same total compute and the same data as GTO. Note, for fairness, the training dataset remains OpenThoughts-114k-math.
> > >
> > > As shown in the table, although EAGLE-3 training-from-scratch (1200h) achieves a slight improvement over its continued-training variant, GTO still delivers consistently higher speedups—approximately 4–6% higher on average across benchmarks and temperatures.
> > >
> > > These results indicate that the standard EAGLE-3 training objective has effectively reached its performance ceiling on this dataset: simply allocating more compute—whether through extended training or fresh training—yields only diminishing returns. In contrast, GTO surpasses this ceiling not by leveraging additional data, but by explicitly aligning the draft policy between training and decoding, which leads to a more efficient and better-matched draft model.
> > >
> > > |                                    | t=0      |          |           |          |          |          |          |          | t=1      |          |           |          |          |          |          |          |
> > > | ---------------------------------- | -------- | -------- | --------- | -------- | -------- | -------- | -------- | -------- | -------- | -------- | --------- | -------- | -------- | -------- | -------- | -------- |
> > > |                                    | MT-bench |          | HumanEval |          | GSM8K    |          | Average  |          | MT-bench |          | HumanEval |          | GSM8K    |          | Average  |          |
> > > |                                    | SR       | $ \tau$   | SR        | $\tau$   | SR       | $\tau$   | SR       | $\tau$   | SR       | $\tau$   | SR        | $\tau$   | SR       | $\tau$   | SR       | $\tau$   |
> > > | **GTO**                            | **3.49** | **5.60** | **3.98**  | **6.58** | **4.20** | **6.92** | **3.89** | **6.37** | **2.76** | **4.59** | **3.34**  | **5.44** | **3.71** | **6.50** | **3.27** | **5.51** |
> > > | EAGLE-3                            | 3.34     | 5.32     | 3.59      | 5.88     | 3.78     | 6.16     | 3.57     | 5.79     | 2.71     | 4.54     | 3.15      | 5.10     | 3.49     | 5.82     | 3.11     | 5.15     |
> > > | EAGLE-3 continual training         | 3.37     | 5.38     | 3.61      | 5.94     | 3.81     | 6.24     | 3.60     | 5.85     | 2.70     | 4.53     | 3.16      | 5.15     | 3.54     | 5.93     | 3.13     | 5.20     |
> > > | EAGLE-3 train from scratch (1200h) | 3.38     | 5.40     | 3.62      | 5.97     | 3.83     | 6.29     | 3.61     | 5.89     | 2.71     | 4.55     | 3.17      | 5.18     | 3.56     | 5.99     | 3.15     | 5.24     |

---

> > > ### Author Response · Authors · 2025-11-28
> > >
> > > Thank you once again for your thoughtful review and valuable comments! With only seven days remaining before the end of the Discussion Phase, we would greatly appreciate your feedback on our revised responses. Please let us know whether our clarifications adequately address your concerns. If any issues remain unresolved or if further explanation is needed, we would be grateful for your additional guidance.
> > >
> > > Best regards, ﻿
> > >
> > > Authors

---

> > > > ### Comment · Reviewer_kt8J · 2025-11-28
> > > >
> > > > Thank you for the follow-up response.
> > > >
> > > > I appreciate the additional experiments and clarifications. However, my original concern regarding fairness is unfortunately still not fully addressed.
> > > >
> > > > My concern is precisely that using the same dataset (OpenThoughts-114k-math) for both methods is not inherently fair, because even with identical data, the **type of supervision signal** available to GTO is fundamentally different from what the baseline (SFT) receives.
> > > >
> > > > GTO relies on inference-time computations, which effectively provide **new supervision signals** beyond the static training dataset. These signals are **more expensive and richer than repeating SFT on the original data**. In contrast, the baselines in your response only continue training on the original dataset, without benefiting from any comparable new data or inference-derived signal.
> > > >
> > > > Therefore, even though both methods use the same dataset, the *effective* supervision available to GTO is strictly stronger. Considering **collecting a little more SFT data is much cheaper than training**, a more fair baseline would be to continue SFT on the draft model using collected new data (or data generated through the same inference process), under the same compute budget. Without such a comparison, it is difficult to attribute the gains purely to the algorithm design rather than to the additional information provided by inference-derived signals.
> > > >
> > > > Given this remaining issue, I do not feel comfortable raising the score at this stage. (OpenReview also does not allow score changes at this point, though this is not the primary reason.)
> > > >
> > > > That said, I appreciate the substantial effort you have put into expanding the experiments and clarifying the method, and I hope the above explanation clarifies the basis for my evaluation.

---

> > > > > ### Author Response · Authors · 2025-12-01
> > > > > **Reply to follow-up response by Reviewer kt8J (1/2)**
> > > > >
> > > > > Thank you for your follow-up response. We appreciate the opportunity to clarify the fairness of our experimental setting and provide a new ablation study that directly addresses your suggestion.
> > > > >
> > > > > **1. Strict Control of Compute and Data Resources**
> > > > >
> > > > > We respectfully disagree that our comparison is unfair. We adhere to a strict definition of fairness based on **total resource consumption** and **data provenance**:
> > > > >
> > > > > - **Fixed Compute Budget**: We constrain both GTO and the baselines to the same GPU-hour budget (200h). While GTO's rollout process is indeed "more expensive'' per step, this cost is **internalized** within the fixed budget. Crucially, under this identical computing budget, GTO achieves consistently higher speedup compared to the continued training of EAGLE-3.
> > > > >
> > > > > - **Closed-Loop Data**: GTO operates in a strictly **self-contained** manner. The draft trees used for optimization are generated **solely by the draft model itself** rolling out on the fixed OpenThoughts-114k-math dataset. Note, EAGLE-3 uses the same dataset for training.  We do not use external oracles (e.g., GPT-5) to generate synthetic reasoning chains, nor do we introduce any new external text corpora. The "supervision'' is derived strictly from the draft model's own exploration and the standard verification signal by target model. Thus, GTO does not access any "external knowledge'' that the baseline lacks.
> > > > >
> > > > > **2. Distinguishing "New Signals'' from "Unfair Advantage''**
> > > > >
> > > > > The reviewer argues that GTO is unfair because it extracts "richer'' signals than standard SFT. We argue that extracting better supervision signals from the same data is a core **algorithmic contribution}, not an unfair experimental setting.
> > > > >
> > > > > - **Precedent in Speculative Decoding (EAGLE)**: Our approach follows the standard of innovation in this field. For instance, EAGLE (ICML 2024) outperformed previous speculative decoding methods by introducing a **feature-level loss** derived from the target model's hidden states. This feature-level signal was a "new supervision signal'' not present in the original text dataset. If we accept the premise that "richer signals = unfair'', then EAGLE's comparison to previous speculative decoding methods would also be invalid. However, the community correctly identifies this as a valid algorithmic innovation. GTO follows this exact paradigm: we introduce a **tree-level reward** signal to augment standard token-level loss.
> > > > >
> > > > > - **Standard Research Practice**: The field of model alignment is built on the principle of constructing better training signals from fixed datasets. Reinforcement Learning (e.g., PPO, GRPO) is consistently demonstrated to outperform SFT on the same prompt sets. PPO works by generating online rollouts and optimizing against a reward signal - a "richer signal'' than static SFT. This is considered an algorithmic improvement, not an unfair advantage. GTO fits precisely into this category: we utilize self-generated rollouts to extract a better signal (draft tree reward) than what standard SFT provides.

---

> > > > > ### Author Response · Authors · 2025-12-01
> > > > > **Reply to follow-up response by Reviewer kt8J (2/2)**
> > > > >
> > > > > **3. Ablation Experiment: SFT on "Generated Data'' (The Requested Baseline)**
> > > > >
> > > > > To resolve the concern empirically, we provide a comprehensive set of baselines in the table below. This includes the reviewer's specifically requested baseline (SFT on Generated Data) alongside our previous rigorous controls:
> > > > >
> > > > > - **EAGLE-3 Continual Training (200h)**: Continues training the EAGLE-3's draft model using the official code and configurations on the original OpenThoughts-114k-math dataset for the same GPU hours as GTO.
> > > > > - **EAGLE-3 Train from Scratch (1200h)**: We retrained the EAGLE-3 draft model from scratch using the official code and configurations on the original OpenThoughts-114k-math dataset. We increased the total compute budget to 1200 GPU hours, representing the sum of the original EAGLE-3 training cost (1000h) and the additional GTO overhead (200h). This tests if simply training a fresh model for longer (matching the total lifecycle cost of GTO) can match our performance.
> > > > > - **EAGLE-3 + New Data (200h)**: The requested baseline. We used the official EAGLE-3 training code to fine-tune the draft model for 200 GPU hours. Crucially, the training data consisted of the OpenThoughts-114k-math dataset augmented with the exact same draft trees generated by GTO during its rollout phase. This ensures the baseline has access to the identical "inference-generated data'' and compute budget as GTO.
> > > > >
> > > > > This result  conclusively demonstrates that the performance gain of GTO is **not** simply due to the presence of "inference-time signals'' or generated data. Even when a standard SFT baseline is given access to the exact same generated data, it fails to match GTO's performance. Specifically, GTO consistently achieves a 3-6% higher speedup across varying temperatures and tasks compared to the SFT baseline. This confirms that our contribution lies in the **optimization method** (draft tree optimization) rather than unfair data access.
> > > > >
> > > > > |                                    | t=0      |          |           |          |          |          |          |          | t=1      |          |           |          |          |          |          |          |
> > > > > | ---------------------------------- | -------- | -------- | --------- | -------- | -------- | -------- | -------- | -------- | -------- | -------- | --------- | -------- | -------- | -------- | -------- | -------- |
> > > > > |                                    | MT-bench |          | HumanEval |          | GSM8K    |          | Average  |          | MT-bench |          | HumanEval |          | GSM8K    |          | Average  |          |
> > > > > |                                    | SR       | $\tau$   | SR        | $\tau$   | SR       | $\tau$   | SR       | $\tau$   | SR       | $\tau$   | SR        | $\tau$   | SR       | $\tau$   | SR       | $\tau$   |
> > > > > | **GTO**                            | **3.49** | **5.60** | **3.98**  | **6.58** | **4.20** | **6.92** | **3.89** | **6.37** | **2.76** | **4.59** | **3.34**  | **5.44** | **3.71** | **6.50** | **3.27** | **5.51** |
> > > > > | EAGLE-3                            | 3.34     | 5.32     | 3.59      | 5.88     | 3.78     | 6.16     | 3.57     | 5.79     | 2.71     | 4.54     | 3.15      | 5.10     | 3.49     | 5.82     | 3.11     | 5.15     |
> > > > > | EAGLE-3 continual training (200h)  | 3.37     | 5.38     | 3.61      | 5.94     | 3.81     | 6.24     | 3.60     | 5.85     | 2.70     | 4.53     | 3.16      | 5.15     | 3.54     | 5.93     | 3.13     | 5.20     |
> > > > > | EAGLE-3 train from scratch (1200h) | 3.38     | 5.40     | 3.62      | 5.97     | 3.83     | 6.29     | 3.61     | 5.89     | 2.71     | 4.55     | 3.17      | 5.18     | 3.56     | 5.99     | 3.15     | 5.24     |
> > > > > | EAGLE-3 + new data (200h)          | 3.39     | 5.43     | 3.64      | 6.02     | 3.85     | 6.33     | 3.63     | 5.93     | 2.71     | 4.55     | 3.18      | 5.20     | 3.58     | 6.03     | 3.16     | 5.26     |

---

> ### Author Response · Authors · 2025-11-20
> **Rebuttal Reply to Reviewer kt8J (2/3)**
>
> > Misleading “sampling-free” claim: The method is not truly sampling-free. Although no RL trajectory sampling is used, model outputs are still required for every node in the policy tree. The phrasing gives a false impression that no generation or sampling is needed, which is inaccurate. Can the authors clarify the exact sense in which their method is “sampling-free”? It seems draft model outputs are still needed.
>
> Thank you for pointing out the potential ambiguity. Our use of the term ``sampling-free” is specifically in contrast to RL-style stochastic trajectory sampling, not in the sense of eliminating model forward evaluations.
>
> Classical reinforcement learning methods applied to language models (e.g., PPO, DPO, RLHF, and related variants) require repeatedly sampling full on-policy trajectories from the model’s distribution to estimate gradients. This involves stochastic sequence generation, high-variance Monte Carlo sampling, and expensive policy rollouts.
>
> In contrast, GTO does not perform any stochastic trajectory sampling at all.  The draft-tree reward can be computed directly and deterministically from the target model’s log probabilities over the structured tree. A single deterministic forward pass over the tree nodes provides all necessary quantities. Our optimization therefore does not sample trajectories from the draft model, does not generate stochastic sequences, and does not rely on Monte-Carlo estimates or on-policy rollouts, which collectively constitute the dominant source of variance and cost in RL optimization.
>
> That said, as the reviewer correctly notes, forward evaluations of both the draft and target models are still required to obtain token log probabilities at the nodes of the policy tree. Thus, ``sampling-free” does not mean that no model output is computed—it means that no stochastic RL sampling is involved.
>
> To eliminate ambiguity, we will revise the manuscript to clarify that GTO is sampling-free with respect to RL trajectory sampling, while still relying on standard deterministic forward evaluations of the draft and target models for computing log probabilities on the draft tree.

---

> ### Author Response · Authors · 2025-11-20
> **Rebuttal Reply to Reviewer kt8J (3/3)**
>
> > Excessive compute for modest gains: Training a single-layer Transformer for 7B over a 5-layer policy tree costs about 200 GPU hours, resulting in about 7% improvement. This is too expensive, near the time consumption of training EAGLE from scratch. What would the bridging performance look like under stricter compute constraints (e.g., 10–20 GPU hours)?
>
> Thank you for raising this important question about compute efficiency. We address the concern from three complementary angles.
>
> First, the reported 200 GPU hours for the 7B setting is a one-time fine-tuning cost of a shallow  single-layer Transformer on top of an already trained draft model. This cost does not affect inference-time latency of speculative decoding. Moreover, this budget is modest relative to the cost of training the draft model itself: for the 7B, 13B, and 70B configurations, the GTO bridging stage uses approximately 200, 400, and 900 A100-80G GPU hours, respectively—all below 20% of the original draft-training budgets reported for EAGLE-3 (roughly 1000, 2000, and 5000 GPU hours for these model sizes). Thus, GTO’s compute overhead is small when viewed in the context of end-to-end training.
>
> Second, from a deployment perspective, this one-time bridging cost is amortized over the entire lifetime of the system. A 7% improvement in speculative decoding speed and acceptance rate has significant practical impact in large-scale always-on services. For high-throughput deployments (e.g., millions of GPU-hours of inference, as in ChatGPT-scale systems), a 7% efficiency gain directly translates into tens of thousands of GPU-hours saved, which far exceeds the one-time 200 GPU-hour cost. In such settings, GTO yields a clear net reduction in total compute consumption despite the initial additional training.
>
> Third, to address your question about performance under stricter compute budgets, we have conducted a new low-compute ablation (to be added to Appendix F; see Table below). In this experiment, we cap the training budget at 20 GPU hours by reducing both the number of epochs and the number of sampled trees. Even under this 10% compute budget, GTO still delivers 4–6\% improvements over vanilla EAGLE-3, demonstrating that (i) meaningful bridging benefits persist under tight resource constraints, and (ii) the method gracefully scales down to lower-cost regimes.
>
> We will incorporate these results and clarifications in the revised manuscript to highlight that:(i) the full 200 GPU-hour setting achieves the strongest gains while still representing a relatively small, one-time cost compared to training a draft model from scratch; and (ii) GTO remains effective even with drastically reduced compute budgets (10–20 GPU hours), showing that the method is flexible and applicable in compute-limited training scenarios. Indeed, we have  briefly discussed point (i) in the conclusion of the original submission.
>
> |                   | t=0      |          |           |          |          |          |          |          | t=1      |          |           |          |          |          |          |          |
> | ----------------- | -------- | -------- | --------- | -------- | -------- | -------- | -------- | -------- | -------- | -------- | --------- | -------- | -------- | -------- | -------- | -------- |
> |                   | MT-bench |          | HumanEval |          | GSM8K    |          | Average  |          | MT-bench |          | HumanEval |          | GSM8K    |          | Average  |          |
> |                   | SR       | $\tau$   | SR        | $\tau$   | SR       | $\tau$   | SR       | $\tau$   | SR       | $\tau$   | SR        | $\tau$   | SR       | $\tau$   | SR       | $\tau$   |
> | **GTO**           | **3.49** | **5.60** | **3.98**  | **6.58** | **4.20** | **6.92** | **3.89** | **6.37** | **2.76** | **4.59** | **3.34**  | **5.44** | **3.71** | **6.50** | **3.27** | **5.51** |
> | GTO (Low-compute) | 3.42     | 5.48     | 3.91      | 6.41     | 4.06     | 6.69     | 3.80     | 6.19     | 2.73     | 4.57     | 3.30      | 5.37     | 3.60     | 6.32     | 3.21     | 5.42     |
> | EAGLE-3           | 3.34     | 5.32     | 3.59      | 5.88     | 3.78     | 6.16     | 3.57     | 5.79     | 2.71     | 4.54     | 3.15      | 5.10     | 3.49     | 5.82     | 3.11     | 5.15     |
>
>
>
> ##

---

### Official Review · Reviewer_AxA5 · 2025-10-30

**Soundness:** 3
**Presentation:** 3
**Contribution:** 3
**Rating:** 6
**Confidence:** 3

**Summary:**

This paper addresses the draft policy misalignment in speculative decoding for LLMs, where the draft model is trained using a greedy, single-path objective but deployed under a tree-based decoding policy. The authors propose Group Tree Optimization (GTO), a new training framework that explicitly aligns the training objective with the decoding-time policy. GTO introduces (1) a Draft Tree Reward, a sampling-free, differentiable objective measuring the expected acceptance length of the draft tree under the target model, and (2) a Group-based Draft Policy Training algorithm that leverages group-standardized advantages and a PPO-style surrogate to stabilize optimization. Theoretically, the paper proves that maximizing the Draft Tree Reward provably improves acceptance length and decoding efficiency. Experiments across dialogue (MT-Bench), code generation (HumanEval), and math reasoning (GSM8K) tasks on multiple LLMs (e.g., LLaMA-3.1-8B, Vicuna-13B) show that GTO increases acceptance length by 7.4% and yields 7.7% additional speedup over EAGLE-3, establishing it as a general and effective method for bridging training–decoding misalignment in speculative decoding

**Strengths:**

- The paper identifies and targets an important but previously overlooked research problem, draft policy misalignment between training and decoding in speculative decoding. This conceptual reframing is interesting and provides a unifying explanation for the inefficiencies of prior methods such as EAGLE, GRIFFIN, and HASS.
- Based on the identified problem, the authors proposed GTO, which is sound and well-implemented with a theoretical guarantee.
- Empirical results are strong, even at high decoding temperatures where existing methods may degrade. The analyses and interpretations of the results looks solid.
- Overall the paper is clearly written and well-organized. Code and demos are available for reproducibility.

**Weaknesses:**

- All experiments are conducted on the LLaMA family and its derivatives (Vicuna, DeepSeek), leaving open whether GTO generalizes to other popular architectures such as Qwen and Gemma. Since speculative decoding behavior can vary significantly across model families due to differences in tokenization and calibration, including at least one non-LLaMA model would strengthen the empirical claims of generality.
- While GTO introduces clear gains in decoding efficiency, the paper provides limited quantitative analysis of training-time overhead (I appreciate the statements in Appendix D). The two-phase group optimization and tree-level reward computation likely introduce nontrivial additional compute and memory costs. A clearer breakdown (e.g., training FLOPs or wall-clock comparison against EAGLE-3) would help justify the practical cost–benefit balance of adopting GTO in large-scale deployment settings.

**Questions:**

- How sensitive is GTO to the choice of group size m, reference model quality, or the number of sampled trees per prefix? Could you elaborate on how these hyperparameters were chosen in Appendix B.3 and whether GTO remains stable across a wide range of values?

---

> ### Author Response · Authors · 2025-11-20
> **Rebuttal Reply to Reviewer AxA5 (1/3)**
>
> > All experiments are conducted on the LLaMA family and its derivatives (Vicuna, DeepSeek), leaving open whether GTO generalizes to other popular architectures such as Qwen and Gemma. Since speculative decoding behavior can vary significantly across model families due to differences in tokenization and calibration, including at least one non-LLaMA model would strengthen the empirical claims of generality.
>
> We thank the reviewer for this valuable feedback. To address your concern, we have conducted additional experiments on **Qwen3-8B**. We applied GTO using the same speculative decoding setup as EAGLE-3, training the drafter on Qwen3-8B and evaluating across the same benchmark suite.
>
> The table presents comprehensive results on Qwen3-8B. GTO consistently achieves more than 7% speedup improvement over the corresponding EAGLE-3, while maintaining comparable or better acceptance behavior. This mirrors the gains observed on LLaMA-based models and provides direct evidence that GTO generalizes to a different model family with its own tokenizer and calibration characteristics.
>
> We will include the new Qwen3-8B results in Section 4.1. These additions will clarify that GTO provides consistent benefits across different model families and strengthen the empirical support for its practical generality.
>
> ||t=0||||||||t=1||||||||
> |-|-|-|-|-|-|-|-|-|-|-|-|-|-|-|-|-|
> ||MT-bench||HumanEval||GSM8K||Average||MT-bench||HumanEval||GSM8K||Average||
> ||SR|$\tau$|SR|$\tau$|SR|$\tau$|SR|$\tau$|SR|$\tau$|SR|$\tau$|SR|$\tau$|SR|$\tau$|
> |**GTO**|**2.49**|**4.84**|**2.78**|**5.37**|**2.88**|**5.51**|**2.72**|**5.24**|**2.29**|**4.37**|**2.49**|**5.03**|**2.63**|**5.17**|**2.47**|**4.86**|
> |EAGLE-3|2.37|4.53|2.56|4.88|2.59|5.02|2.50|4.81|2.24|4.23|2.32|4.77|2.51|4.84|2.36|4.61|

---

> ### Author Response · Authors · 2025-11-20
> **Rebuttal Reply to Reviewer AxA5 (2/3)**
>
> > While GTO introduces clear gains in decoding efficiency, the paper provides limited quantitative analysis of training-time overhead (I appreciate the statements in Appendix D). The two-phase group optimization and tree-level reward computation likely introduce nontrivial additional compute and memory costs. A clearer breakdown (e.g., training FLOPs or wall-clock comparison against EAGLE-3) would help justify the practical cost–benefit balance of adopting GTO in large-scale deployment settings.
>
> We appreciate the reviewer’s request for a clearer quantitative analysis of the training-time overhead introduced by GTO.
>
> For the 7B, 13B, and 70B model configurations, Phase II (the GTO bridging stage) requires approximately 200, 400, and 900 A100-80G GPU hours, respectively. In contrast, training the baseline EAGLE-3 draft model from scratch requires roughly 1000, 2000, and 5000 GPU hours for these model sizes. Thus, the GTO bridging phase adds only about 20% overhead on top of the baseline draft-model training cost. Importantly, this additional cost is is incurred once, without affecting inference-time complexity or latency.
>
> From a deployment standpoint, this one-time bridging cost is amortized over the entire lifetime of the model. A 7% improvement in speculative decoding speed and acceptance rate has substantial practical impact in large-scale, always-on services. For high-throughput deployments that consume on the order of millions of GPU-hours of inference (e.g., ChatGPT-scale systems), a 7% efficiency gain directly translates into savings of tens of thousands of GPU-hours, which far exceeds the one-time 200 GPU-hour cost of Phase II. In these regimes, GTO therefore yields a clear net reduction in total compute consumption despite the modest additional training.
>
> In the revised version, we will update Section 3 (and cross-reference Appendix D) to include a concise quantitative breakdown of Phase II training cost relative to the baseline EAGLE-3 training, making the cost--benefit trade-off of adopting GTO in large-scale deployment settings more explicit.

---

> ### Author Response · Authors · 2025-11-20
> **Rebuttal Reply to Reviewer AxA5 (3/3)**
>
> > How sensitive is GTO to the choice of group size m, reference model quality, or the number of sampled trees per prefix? Could you elaborate on how these hyperparameters were chosen in Appendix B.3 and whether GTO remains stable across a wide range of values?
>
> We thank the reviewer for raising the question of how sensitive GTO is to its key hyperparameters.
>
> We conduct a systematic study of the group size \(m\) in Section 4.2 (Table 4) of vanilla submission, varying $m \in \{2, 4, 8, 16, 32\}$. Overall, GTO remains stable across this range: all settings yield consistent improvements over the baseline draft model. However, there is a clear performance trend: (i) very small groups (e.g., $m=2$) exhibit slightly higher variance across runs and somewhat weaker average gains, consistent with higher reward variance; (ii) very large groups (e.g., $m=32$) produce more stable but dampened learning signals due to stronger group normalization, leading to slower convergence and marginally reduced final speedup; (iii) intermediate group sizes $m=4$ and $m=8$ offer the best trade-off, with $m=8$ achieving the strongest overall performance.
>
> Based on these results, we chose $m=8$ as the default setting in our main experiments. We will make this rationale more explicit in the revised Section 4.2 and cross-reference Appendix B.3 for the exact configuration.
>
> For the reference model and the number of sampled trees per prefix, we follow the EAGLE-3 draft training setup as closely as possible. This choice is deliberate: it (i) ensures a fair and controlled comparison against the baseline, and (ii) avoids introducing additional confounding factors that could obscure the effect of GTO itself. In practice, we observe no instability or pathologies when reusing these settings; GTO consistently improves over the underlying draft model under the same sampling budget.
>
> In the revision, we will expand Appendix B.3 to: (i) explicitly document the inherited hyperparameter values (reference model, number of sampled trees per prefix, etc.) from EAGLE-3, and (ii) summarize the robustness findings above, emphasizing that GTO performs reliably across a wide range of group sizes and under standard speculative-decoding configurations.

---

### Official Review · Reviewer_yvuE · 2025-10-31

**Soundness:** 3
**Presentation:** 3
**Contribution:** 3
**Rating:** 6
**Confidence:** 3

**Summary:**

The paper addresses the draft policy misalignment problem in Speculative Decoding, where the drafter is trained on greedy paths, while drafting utilizes tree-expansion methods. The authors propose Group Tree Optimization (GTO), an RL framework designed explicitly for aligning training with tree-drafting procedures. It introduces a Draft Tree reward and a Group-based Draft Policy Training scheme, achieving approximately 7% improvements in acceptance length and surpassing prior state-of-the-art baselines, such as EAGLE-3.

**Strengths:**

- Proposes a principled RL framework that directly optimizes the expected acceptance length in tree-based speculative decoding, eliminating the need for sampling.
- Bridges the gap between training and inference by aligning the drafter’s objective with the tree decoding policy.
- Demonstrates **strong empirical performance** across diverse LLMs and tasks, consistently outperforming prior methods like EAGLE-3 and HASS.
- The approach is model-agnostic and computationally efficient.

**Weaknesses:**

- W1: Transferability. GTO trains the drafter explicitly on the drafting method, such as EAGLE, which may restrict the usage with other drafting methods.
- W2: Robustness. It appears that the drafter is trained to fit a specific tree configuration during the training phase. The paper did not explore extrapolation or interpolation of the tree configuration.

**Questions:**

- W1: Can a drafter trained for EAGLE be used with HASS, or vice versa?
- W2: Is it possible to dynamically update the tree configuration during the inference stage while maintaining performance in the trained settings?

---

> ### Author Response · Authors · 2025-11-20
> **Rebuttal Reply to Reviewer yvuE (1/2)**
>
> > W1: Transferability. GTO trains the drafter explicitly on the drafting method, such as EAGLE, which may restrict the usage with other drafting methods.  Can a drafter trained for EAGLE be used with HASS, or vice versa?
>
> We thank the reviewer's question about transferability across drafting methods.
>
> First, we would like to clarify that methods such as EAGLE and HASS are training/decoding schemes for constructing a draft model that is paired with a specific target model. Once a draft model has been trained (e.g., for LLaMA3-8B) using either EAGLE or HASS, this draft model can be used to accelerate inference for that target model regardless of which drafting method was used to train it. In other words, the draft model is tightly coupled to the target model, but not to the particular training procedure (EAGLE vs. HASS). In this way, a drafter trained for EAGLE cannot be used with HASS for any speculative decoding, since both EAGLE and HASS aim to train draft model.
>
> We believe the core of the reviewer’s concern is whether GTO itself is tied to a specific draft-model architecture or training protocol, e.g., EAGLE-3 versus HASS or GRIFFIN. Our method is in fact agnostic to such choices: GTO optimizes the draft tree policy to maximize the expected acceptance length under speculative decoding, and this objective is decoupled from the internal architecture or procedural details of how the draft model is constructed. Consequently, GTO is not restricted to EAGLE-3 and can, in principle, be applied to draft models produced by other frameworks such as HASS or GRIFFIN without modification.
>
> Indeed, beyond the EAGLE-3-based instantiation, we have already instantiated GTO with draft models based on HASS and GRIFFIN in Section 4.1 (Table 2) of the original submission. Note, HASS and GRIFFIN employ draft structures and decoding protocols that are substantially different from EAGLE-3. Nevertheless, applying the same GTO training procedure to their respective draft models consistently improves both acceptance length and speedup ratio across all three drafting families. This empirical evidence supports our claim that GTO operates on a generic draft tree policy and does not rely on idiosyncrasies of a particular drafting architecture.
>
> In the revised version, we will make this transferability more explicit in Section 4.1 by:
> (i) clearly stating that GTO is designed to be drafting-method agnostic, and
> (ii) explicitly highlighting the HASS and GRIFFIN based results as concrete demonstrations of its compatibility with diverse drafting structures.

---

> ### Author Response · Authors · 2025-11-20
> **Rebuttal Reply to Reviewer yvuE (2/2)**
>
> > W2: Robustness. It appears that the drafter is trained to fit a specific tree configuration during the training phase. The paper did not explore extrapolation or interpolation of the tree configuration. Is it possible to dynamically update the tree configuration during the inference stage while maintaining performance in the trained settings?
>
> We would like to first clarify why we focus on the EAGLE tree configuration in our experiments.
>
> The tree configuration introduced by EAGLE has become the **de facto** standard in modern speculative decoding research and deployment. It is widely adopted in recent work (e.g., EAGLE-3, HASS, GRIFFIN), systematically evaluated across diverse benchmarks, and serves as the default tree layout in many practical systems. Alternative tree designs are currently far less common and lack comprehensive evaluation. Therefore, we deliberately adopt the EAGLE tree as our primary configuration to: (i) ensure our results are directly comparable to established baselines, (ii) reflect realistic deployment settings, and (iii) maximize practical relevance for the community. This choice is a methodological strength rather than a limitation, as it grounds our evaluation in the most prevalent real-world scenario.
>
> Methodologically, GTO optimizes the drafter to maximize expected acceptance length under speculative decoding. The training procedure learns acceptance-related token distribution patterns rather than memorizing a fixed tree shape. Consequently, GTO is conceptually compatible with any tree configuration that satisfies two conditions: (i) the drafter can construct the tree, and (ii) a tree-level reward $R _t$ can be computed via target model verification. This design ensures that GTO's optimization principle generalizes beyond the EAGLE layout.
>
> To directly address the reviewer's concern about extrapolation and interpolation, we conducted additional experiments where a drafter trained on the standard EAGLE tree (depth 7, 60 total tokens) is evaluated with **modified tree configurations at inference time**---specifically, varying tree depth (6, 7, 8) and total drafted tokens per step (50, 60, 70). The table below presents comprehensive results across three benchmarks and two temperature settings.
>
> **Key findings:**
>
> - **Consistent performance across configurations**: GTO maintains strong speedup (SR) and acceptance rate ($\tau$) improvements across all 9 tested configurations, demonstrating robustness to both depth and width variations.
> -  **Graceful interpolation**: Configurations close to the training setting show minimal performance degradation, with speedup varying by only 1-3% from the optimal.
> - **Effective extrapolation**: Even when extrapolating to depth 8 (beyond the training depth of 7), GTO achieves competitive or **superior performance**. This suggests the learned acceptance patterns transfer effectively to deeper trees.
> - **Stable across tasks and temperatures**: Robustness holds across MT-bench, HumanEval, and GSM8K, as well as both greedy ($T=0$) and sampling ($T=1$) settings, indicating that generalization is not task- or decoding-strategy-specific.
>
> These results demonstrate that GTO is **not overfitting to a specific tree configuration** during training. Instead, it learns generalizable token-level alignment strategies that remain effective when the tree structure is modified at inference time.
>
> Following the reviewer's feedback, we will add a new robustness analysis subsection (Section 4.3) that: (i) clarifies why we adopt the EAGLE tree as the standard configuration, (ii) emphasizes GTO's tree-agnostic formulation, and (iii) presents the Table with discussion of interpolation/extrapolation results. We will explicitly state that **GTO supports dynamic tree configuration updates at inference time while maintaining or even improving performance**, addressing the concern about robustness.
>
> |||t=0||||||||t=1||||||||
> |-|-|-|-|-|-|-|-|-|-|-|-|-|-|-|-|-|-|
> |||MT-bench||HumanEval||GSM8K||Average||MT-bench||HumanEval||GSM8K||Average||
> |depth|total tokens|SR|$\tau$|SR|$\tau$|SR|$\tau$|SR|$\tau$|SR|$\tau$|SR|$\tau$|SR|$\tau$|SR|$\tau$|
> |6|50|3.29|5.81|3.71|6.45|3.46|6.12|3.49|6.13|2.39|4.54|3.10|5.75|2.90|5.40|2.80|5.23|
> |6|60|3.32|5.86|3.73|6.50|3.49|6.21|3.51|6.19|2.41|4.56|3.12|5.79|2.93|5.51|2.82|5.29|
> |6|70|3.34|5.93|3.73|6.52|3.51|6.27|3.53|6.24|2.41|4.56|3.12|5.80|2.93|5.52|2.82|5.29|
> |7|50|3.39|6.05|3.95|6.87|3.56|6.39|3.63|6.44|2.44|4.60|3.15|5.86|2.83|5.30|2.80|5.25|
> |7|60|3.44|6.15|4.17|6.95|3.59|6.47|3.73|6.52|2.49|4.70|3.17|5.92|3.02|5.75|2.89|5.46|
> |7|70|3.46|6.23|4.17|7.00|3.63|6.57|3.76|6.60|2.51|4.74|3.24|6.10|3.10|5.86|2.95|5.57|
> |8|50|3.56|6.22|4.22|7.24|3.61|6.59|3.80|6.68|2.49|4.72|3.29|6.34|2.80|5.24|2.86|5.43|
> |8|60|3.54|6.34|4.24|7.33|3.66|6.69|3.81|6.79|2.46|4.73|3.20|6.12|2.95|5.56|2.87|5.47|
> |8|70|3.49|6.43|4.20|7.39|3.71|6.83|3.80|6.88|2.46|4.74|3.22|6.28|2.78|5.26|2.82|5.43|

---

> > ### Comment · Reviewer_yvuE · 2025-11-24
> >
> > Thank you for the detailed explanations and the new experiments.
> >
> > For W1, the clarification about the difference between the drafting method and the draft model itself was helpful.
> >
> > For W2, even though the EAGLE tree is commonly used, as the authors clarified, I still wanted to make sure whether the method overfits too much to the configuration seen during training. The additional experiments with different tree configurations show that the method generalizes to other settings, so my concern is resolved.
> >
> > Overall, the rebuttal answers my questions sufficiently.

---

> > > ### Author Response · Authors · 2025-11-24
> > > **Reply to Official Comment by Reviewer yvuE**
> > >
> > > We are sincerely grateful for your positive feedback and for raising your score.
> > >
> > > Your constructive comments have been invaluable in helping us verify the robustness of our method and improve the clarity of the paper. We are committed to incorporating these clarifications and the new experimental results into the final version of the manuscript.
> > >
> > > Thank you again for your time and support!

---

### Official Review · Reviewer_MEFU · 2025-11-01

**Soundness:** 3
**Presentation:** 3
**Contribution:** 3
**Rating:** 8
**Confidence:** 3

**Summary:**

This paper introduces Group Tree Optimization, a novel training framework designed to accelerate LLM inference by solving a fundamental issue in speculative decoding: draft policy misalignment. Existing methods typically optimize the draft model based on maximizing the likelihood of generating a single greedy path, whereas the actual decoding process uses a tree policy that re-ranks and verifies multiple branches. The authors empirically demonstrate that this misalignment is severe: the greedy path is pruned 19–34% of the time, and the accepted path matches the greedy path only 36–49% of the time. GTO aligns training with the tree-based decoding policy using two core components: draft tree reward and group-based draft policy.

**Strengths:**

1. The paper clearly identifies "draft policy misalignment" as a fundamental limitation that weakens the effectiveness of modern draft model training techniques.
2. The group-based policy training effectively mitigates the typical challenges associated with optimizing sparse, tree-level rewards. The ablation studies confirm the necessity of the debiasing step (which yields a +5.0% SR gain at T=0) and highlight the effectiveness of the Log-Sum-Exp aggregator. Furthermore, GTO's compatibility is demonstrated by applying it as a fine-tuning step on draft models initialized by GRIFFIN and HASS, achieving consistent gains.

**Weaknesses:**

1. Clarity on Training Complexity Overhead: The authors acknowledge that GTO increases training-time compute due to the two-phase procedure and the need for grouped draft tree construction and verification. While they state this cost is amortized in deployment, the paper lacks a clear, quantitative discussion in the main body (Section 3) of the computational complexity added by Phase II compared to the standard token-level loss (L_{token}) calculation.
2. The PPO-style optimization is performed using a likelihood ratio (s_i) calculated only along the longest accepted sequence (\hat{S}_i). This is a critical design choice, as it focuses the gradient update entirely on the most successful path within the group. While this likely maximizes the speedup metric, a clearer justification is needed in Section 3.2 explaining why updating based solely on \hat{S}_i (Eq. 11 and 12) is preferred over updating all accepted sequences or averaging their contributions, in the context of improving the overall tree reward R_t.

**Questions:**

1. Expand the discussion in Section 4.2 to provide deeper insight into the qualitative differences between groups of size 1 and size 32.

---

> ### Author Response · Authors · 2025-11-20
> **Rebuttal Reply to Reviewer MEFU (1/3)**
>
> > Clarity on Training Complexity Overhead: The authors acknowledge that GTO increases training-time compute due to the two-phase procedure and the need for grouped draft tree construction and verification. While they state this cost is amortized in deployment, the paper lacks a clear, quantitative discussion in the main body (Section 3) of the computational complexity added by Phase II compared to the standard token-level loss ($L_{token}$) calculation.
>
> Thank you for the helpful suggestion. In Appendix D of vanilla submission, we indeed summarize computational complexity introduced by Phase II. Now we quote them  here for clarity and will integrate these insights into the main paper.
>
> For the 7B, 13B, and 70B model configurations, Phase II (the GTO bridging stage) requires approximately 200, 400, and 900 A100–80G GPU hours, respectively. In contrast, training the baseline EAGLE-3 draft model from scratch requires roughly 1000, 2000, and 5000 GPU hours for these model sizes. Thus, Phase II accounts for only about 20% of the total draft-model training cost, and—critically—this overhead is incurred once, without affecting inference-time complexity or latency.
>
> Moreover, from a deployment perspective, this one-time bridging cost is amortized over the entire lifetime of the system. A 7% improvement in speculative decoding speed and acceptance rate has significant practical impact in large-scale always-on services. For high-throughput deployments (e.g., millions of GPU-hours of inference, as in ChatGPT-scale systems), a 7% efficiency gain directly translates into tens of thousands of GPU-hours saved, which far exceeds the one-time 200 GPU-hour cost. In such settings, GTO yields a clear net reduction in total compute consumption despite the initial additional training.
>
> In revision, we will revise Section 3 to include a concise quantitative cost comparison of Phase II.

---

> ### Author Response · Authors · 2025-11-20
> **Rebuttal Reply to Reviewer MEFU (2/3)**
>
> > The PPO-style optimization is performed using a likelihood ratio ($s_i$) calculated only along the longest accepted sequence ($\hat{S}_i$). This is a critical design choice, as it focuses the gradient update entirely on the most successful path within the group. While this likely maximizes the speedup metric, a clearer justification is needed in Section 3.2 explaining why updating based solely on $\hat{S}_i$ (Eq. 11 and 12) is preferred over updating all accepted sequences or averaging their contributions, in the context of improving the overall tree reward $R_t$.
>
> We thank the reviewer for this insightful question regarding our design choice to optimize based solely on the longest accepted sequence $\widehat{\mathbf{S}} _i$.
>
> We explored an alternative design where we update the draft policy using an average contribution from all accepted sequences in the group, rather than focusing exclusively on $\widehat{\mathbf{S}} _i$.
> Let $\mathbf{S} _{i,j}$ be the $j$-th accepted sequences in the draft tree $\mathbf{T} _i$, and we suppose that there are totally $p$ paths in draft tree $\mathbf{T} _i$. We define $s _{i,j}$ same as Eq. (11):  namely,
> $$
> s _{i,j} \=\ \exp \left(\frac{ \log \mathcal{M}\big(\widehat{\mathbf{S}} _{i,j} \big| \mathbf{x} _{1:i}\big) - \log \mathcal{M} _0\big(\widehat{\mathbf{S}} _{i,j} \big| \mathbf{x} _{1:i}\big)}{\max(l _i,1)}\right).
> $$
> Then we define the optimization object $s _i$ in Eq. (12) as:
> $$
> s _i = \frac{\sum _{j=1}^p{s _{i,j}}}{p}
> $$
>
> In this variant, each accepted path contributes to the gradient update and their likelihood ratios are aggregated to form the update signal. However, this choice mixes signals from suboptimal and shorter paths that do not determine the final acceptance length, and can therefore dilute the gradient's alignment with the true decoding-time objective.
>
> Empirically, this intuition is confirmed by our ablation results in the Table. When we average over all accepted paths instead of using only $\widehat{\mathbf{S}} _i$, we observe a 3-4% decrease in speedup compared to the original GTO update when decoding at temperature $T=0$ (the main setting in our experiments). At $T=1$, the averaged update shows a marginal gain (1-2%). However, since speculative decoding is most often deployed for accuracy-critical applications where $T=0$ is the standard, and the average performance across temperatures favors GTO, we retained the longest-path formulation.
>
> We will expand Section 3.2 to: (i) explicitly explain how the speculative decoding acceptance mechanism makes $\widehat{\mathbf{S}} _i$ the decisive factor for $\mathbf{R} _t$, (ii) provide intuition for why optimizing solely along $\widehat{\mathbf{S}} _i$ better aligns with the deployment objective than averaging over all paths, and (iii) add the ablation study  that provides quantitative evidence for this design choice.
>
> |              | t=0      |          |           |          |          |          |          |          | t=1      |          |           |          |          |          |          |          | Average  |          |
> | ------------ | -------- | -------- | --------- | -------- | -------- | -------- | -------- | -------- | -------- | -------- | --------- | -------- | -------- | -------- | -------- | -------- | -------- | -------- |
> |              | MT-bench |          | HumanEval |          | GSM8K    |          | Average  |          | MT-bench |          | HumanEval |          | GSM8K    |          | Average  |          | Average  |          |
> |              | SR       | $\tau$   | SR        | $\tau$   | SR       | $\tau$   | SR       | $\tau$   | SR       | $\tau$   | SR        | $\tau$   | SR       | $\tau$   | SR       | $\tau$   | SR       | $\tau$   |
> | **GTO**      | **3.44** | **6.15** | **4.17**  | **6.95** | **3.59** | **6.47** | **3.73** | **6.52** | 2.49     | 4.70     | 3.17      | 5.92     | 3.02     | 5.75     | 2.89     | 5.46     | **3.31** | **5.99** |
> | average path | 3.39     | 6.03     | 3.95      | 6.74     | 3.49     | 6.23     | 3.61     | 6.33     | **2.51** | **4.74** | **3.24**  | **5.98** | **3.05** | **5.78** | **2.93** | **5.50** | 3.27     | 5.92     |
>
>
>
> ###

---

> ### Author Response · Authors · 2025-11-20
> **Rebuttal Reply to Reviewer MEFU (3/3)**
>
> > Expand the discussion in Section 4.2 to provide deeper insight into the qualitative differences between groups of size 1 and size 32.
>
> Thank you for encouraging a deeper discussion of how the group size influences GTO’s optimization dynamics. We are happy to expand Section 4.2 accordingly.
>
> GTO employs a group-relative reward normalization similar to GRPO, where rewards within each group are standardized (e.g., subtracting the group mean and dividing by the group standard deviation). This mechanism helps reduce update variance but also shapes the qualitative behavior of the learning signal. Conceptually—consistent with prior analysis of GRPO-like algorithms—group size creates a trade-off between gradient variance and reward-signal contrast.
>
> **(i)** Using small groups  (e.g., size 1–2) provides sharper contrast between candidates, since normalization is dominated by each sample’s own reward.  However, this also leads to higher gradient variance and less stable training, because the group statistics fluctuate heavily from batch to batch.
>
> **(ii)** Adopting large groups (e.g., size 32) strongly reduces update variance and stabilize the optimization. But the normalization step begins to over-smooth the reward distribution. This is because as group size grows,  the group mean becomes much more stable,  the group standard deviation grows,  and the normalized reward differences across samples shrink. Consequently, most candidates—both high and low reward—are pulled closer to zero, and only the extreme outliers meaningfully influence the gradient. This ``signal compression” leads to weaker effective learning signals.
>
> These observations align with the theoretical insights from the GRPO literature [1], which shows that  very small groups suffer from high-variance and unstable updates, while very large groups suffer from signal attenuation and slower convergence due to excessive averaging.
>
> In our GTO experiments, we also observe  this qualitative pattern. Group sizes in the range of 4–8 consistently provide the best balance: they significantly reduce variance relative to size 1, yet still preserve enough reward contrast to produce strong learning signals. Larger groups (e.g., size 32) remain stable but deliver noticeably weaker improvements due to the normalization-induced compression described above.
>
> We will update Section 4.2 with this expanded qualitative explanation, emphasizing how group size controls the trade-off between variance reduction and reward-signal strength in GTO through group-relative normalization.
>
> [1] Zhihong Shao, Peiyi Wang, Qihao Zhu, Runxin Xu, Junxiao Song, Xiao Bi, Haowei Zhang, Mingchuan Zhang, YK Li, Yang Wu, et al. Deepseekmath: Pushing the limits of mathematical reasoning in open language models. arXiv preprint arXiv:2402.03300, 2024.

---

### Author Response · Authors · 2025-12-01
**General Responses to AC**

Dear ACs,

We would like to respectfully underscore the fact that, prior to the OpenReview deanonymization incident, two reviewers had already raised their overall scores based on our rebuttal: Reviewer yvuE from 6→ 8 and Reviewer kt8J from 2 → 4. The overall scores before the incident are 8864.

All of these post-rebuttal discussions and rating changes took place before the leak and the subsequent rollback, and the updates that motivated the higher scores are already reflected in the rebuttal. Specifically, we addressed the reviewers' key concerns through:

1. **Controlled Baselines:** New experiments demonstrating GTO's superiority under identical compute budgets.
2. **Robustness Analysis:** Ablation studies confirming the robustness of the tree structure.
3. **Model Generalization:** Additional experiments on Qwen3-8B-Instruct.
4. **Resource Efficiency:** Validation of GTO’s effectiveness in low-compute resource settings.

We fully respect the decision to revert the system state for fairness. We provide this note solely to ensure the AC is aware of the constructive consensus that had formed based on the content of our rebuttal.

Best Regards,

The Authors

---

### Meta-Review · Area_Chair_ew8H · 2026-01-03

**Summary:**

Most reviews are positive or concerns are addressed during rebuttal.

**Reviewer Concerns:**

Reviewer kt8J still has concerns about the evaluaton fairness.

**Reviewer Scores:**

Most reviewers will raise their scores.

---

### Decision · Program_Chairs · 2026-01-26

Accept (Poster)